# Variational methods for simulation-based inference

**Manuel Glöckler**
University of Tübingen

**Michael Deistler**
University of Tübingen

**Jakob H. Macke**
University of Tübingen

## Abstract

We present Sequential Neural Variational Inference (SNVI), an approach to perform Bayesian inference in models with intractable likelihoods. SNVI combines likelihood-estimation (or likelihood-ratio-estimation) with variational inference to achieve a scalable simulation-based inference approach. SNVI maintains the flexibility of likelihood(-ratio) estimation to allow arbitrary proposals for simulations, while simultaneously providing a functional estimate of the posterior distribution without requiring MCMC sampling. We present several variants of SNVI and demonstrate that they are substantially more computationally efficient than previous algorithms, without loss of accuracy on benchmark tasks. We apply SNVI to a neuroscience model of the pyloric network in the crab and demonstrate that it can infer the posterior distribution with one order of magnitude fewer simulations than previously reported. SNVI vastly reduces the computational cost of simulation-based inference while maintaining accuracy and flexibility, making it possible to tackle problems that were previously inaccessible.

## 1 Introduction

Many domains in science and engineering use numerical simulations to model empirically observed phenomena. These models are designed by domain experts and are built to produce mechanistic insights. However, in many cases, some parameters of the simulator cannot be experimentally measured and need to be inferred from data. A principled way to identify parameters that match empirical observations is Bayesian inference. However, for many models of interest, one can only *sample* from the model by simulating a (stochastic) computer program, but explicitly evaluating the likelihood $p(\boldsymbol{x}|\boldsymbol{\theta})$ is intractable. Traditional methods to perform Bayesian inference in such *simulation-based inference* (SBI), also known as *likelihood-free* inference scenarios, include Approximate Bayesian computation (ABC) (Beaumont et al., 2002) and synthetic likelihood (SL) (Wood, 2010) methods. However, these methods generally struggle with high-dimensional data and typically require one to design or learn (Chen et al., 2021) summary statistics and distance functions.

Recently, several methods using neural density(-ratio) estimation have emerged. These methods train neural networks to learn the posterior (SNPE, Papamakarios & Murray, 2016; Lueckmann et al., 2017; Greenberg et al., 2019), the likelihood (SNLE, Papamakarios et al., 2019; Lueckmann et al., 2019a), or the likelihood-to-evidence ratio (SNRE, Thomas et al., 2021; Hermans et al., 2020; Durkan et al., 2020; Miller et al., 2022).

To improve the simulation efficiency of these methods, sequential training schemes have been proposed: Initially, parameters are sampled from the prior distribution to train an estimation-network. Subsequently, new samples are drawn adaptively to focus training on specific regions in parameter space, thus allowing the methods to scale to larger models with more parameters.

In practice, however, it has remained a challenge to realize the full potential of these sequential schemes: For sequential neural posterior estimation (SNPE) techniques, the loss function needs to be adjusted across rounds (Greenberg et al., 2019), and it has been reported that this can be problematic if the proposal distribution is very different from prior, and lead to 'leakage' of probability mass into regions without prior support (Durkan et al., 2020). Both sequential neural likelihood (SNLE) and likelihood-ratio (SNRE) methods require MCMC sampling, which can become prohibitively slow–MCMC sampling is required for each round of simulations, which, for high-dimensional models, can take more time than running the simulations and training the neural density estimator.

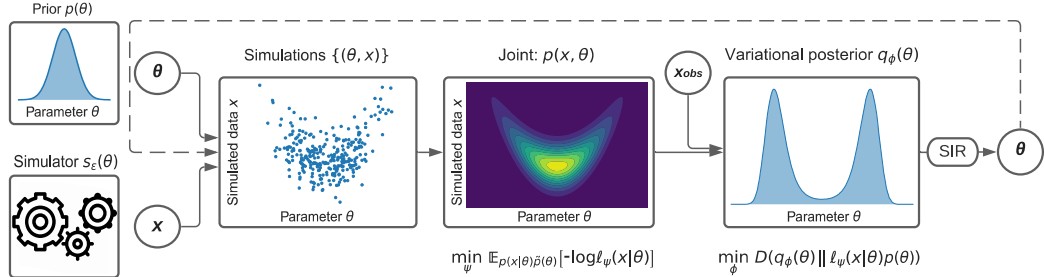

Figure 1: Illustration of SNVI. We first learn the likelihood $p(\boldsymbol{x}|\boldsymbol{\theta})$ for any $\boldsymbol{\theta}$. We then use variational inference to learn the posterior distribution by minimizing a general divergence measure $D$. The obtained posterior distribution is sampled with sampling importance resampling (SIR) to run new simulations and refine the likelihood estimator.

Our goal is to provide a method which combines the advantages of posterior-targeting methods and those targeting likelihood(-ratios): Posterior targeting methods allow rapid inference by providing a functional approximation to the posterior which can be evaluated without the need to use MCMC sampling. Conversely, a key advantage of likelihood(-ratio) targeting methods is their flexibility– learned likelihoods can e.g. be used to integrate information from multiple observations, or can be used without retraining if the prior is changed. In addition, they can be applied with any active-learning scheme without requiring modifications of the loss-function.

We achieve this method by combining likelihood(-ratio) estimation with variationally learned inference networks using normalizing flows (Rezende & Mohamed, 2015; Papamakarios et al., 2017; Durkan et al., 2019a) and sampling importance resampling (SIR) (Rubin, 1988). We name our approach Sequential Neural Variational Inference (SNVI). We will show that our simulation-based inference methods are as accurate as SNLE and SNRE, while being substantially faster at inference as they do not require MCMC sampling. In addition, real-world simulators sometimes produce invalid outputs, e.g. when a simulation fails. We introduce a strategy that allows likelihood(-ratio) targeting methods (such as SNVI) to deal with such invalid simulation outputs.

A recent method termed "Sequential Neural Posterior and Likelihood Approximation" (SNPLA) also proposed to use variational inference (VI) instead of MCMC to speed up inference in likelihood-targeting methods (Wiqvist et al., 2021). While this proposal is related to our approach, their VI objective is based on the reverse Kullback Leibler (rKL) divergence for learning the posterior. As we also show on benchmark tasks, this leads to mode-seeking behaviour which can limit its performance. In contrast, we show how this limitation can be overcome through modifying the variational objective in combination with using SIR for adjusting posteriors.

After an introduction on neural network-based simulation-based inference (SBI) and variational inference (Sec. 2), we present our method, Sequential Neural Variational Inference (SNVI) (Sec. 3). In Sec. 4.2, we empirically show that SNVI is significantly faster than state-of-the-art SBI methods while achieving similar accuracy on benchmark tasks. In Sec. 4.3, we demonstrate that SNVI is scalable, and that it is robust to invalid simulation outputs: We obtain the posterior distribution of a complex neuroscience model with one order of magnitude fewer simulations than previous methods.

## 2 BACKGROUND

### 2.1 SIMULATION-BASED INFERENCE

Simulation-based inference (SBI) aims to perform Bayesian inference on statistical models for which the likelihood function is only implicitly defined through a stochastic simulator. Given a prior $p(\boldsymbol{\theta})$ and a simulator which implicitly defines the likelihood $p(\boldsymbol{x}|\boldsymbol{\theta})$, the goal is to identify the posterior distribution $p(\boldsymbol{\theta}|\boldsymbol{x}_o)$ for an observation $\boldsymbol{x}_o$. The simulator is considered to be 'black-box', i.e. one cannot evaluate $p(\boldsymbol{x}|\boldsymbol{\theta})$ and does not have access to the internal states of the simulator, but only to its inputs $\boldsymbol{\theta}$ and its outputs $\boldsymbol{x}$.

We focus on improving likelihood-estimation (SNLE) and likelihood-ratio-estimation (SNRE) methods. SNLE trains a deep neural density estimator $\ell_\psi(\boldsymbol{x}|\boldsymbol{\theta})$ by minimizing the forward Kullback-Leibler divergence (fKL) between $\ell_\psi(\boldsymbol{x}|\boldsymbol{\theta})$ and $p(\boldsymbol{x}|\boldsymbol{\theta})$ using samples $(\boldsymbol{x},\boldsymbol{\theta}) \sim \tilde{p}(\boldsymbol{x},\boldsymbol{\theta}) = p(\boldsymbol{x}|\boldsymbol{\theta})\tilde{p}(\boldsymbol{\theta})$ from the simulator with $\mathcal{L}(\psi) = -\frac{1}{N}\sum_{i=1}^{N}\log\ell_\psi(\boldsymbol{x}_i|\boldsymbol{\theta}_i)$. Here, $\ell_\psi(\boldsymbol{x}|\boldsymbol{\theta})$ is a conditional density estimator learning the conditional density $p(\boldsymbol{x}|\boldsymbol{\theta})$ from $(\boldsymbol{\theta},\boldsymbol{x})$ pairs, $\psi$ are its learnable parameters, and $\tilde{p}(\boldsymbol{\theta})$ is the proposal distribution from which the parameters $\boldsymbol{\theta}$ are drawn (given by e.g. a previous estimate of the posterior or by an active learning scheme, Papamakarios et al., 2019; Lueckmann et al., 2019a).

Analogously, SNRE uses a discriminator, e.g. a deep logistic regression network, to estimate the density *ratio* $r(\boldsymbol{x},\boldsymbol{\theta}) = \frac{\tilde{p}(\boldsymbol{x},\boldsymbol{\theta})}{\tilde{p}(\boldsymbol{x})\tilde{p}(\boldsymbol{\theta})} = \frac{p(\boldsymbol{x}|\boldsymbol{\theta})}{\tilde{p}(\boldsymbol{x})}$. (Hermans et al., 2020; Durkan et al., 2020), If the proposal is given by the prior, then one can recover the exact posterior density, otherwise the posterior can be recovered up to a normalizing constant (Durkan et al., 2020). Once the likelihood (or likelihood-ratio) has been learned, the posterior can be sampled with MCMC. In sequential schemes, the proposal $\tilde{p}$ is updated each round using the current estimate of the posterior – thus, computationally expensive MCMC sampling needs to be run in each round.

## 2.2 VARIATIONAL INFERENCE

We use variational inference (VI) to estimate the posterior distribution. VI formulates an optimization problem over a class of tractable distributions $\mathcal{Q}$ to find parameters $\phi^*$ such that $q_{\phi^*} \in \mathcal{Q}$ is closest to the true posterior $p(\boldsymbol{\theta}|\boldsymbol{x}_o)$ according to some divergence $D$ (Blei et al., 2017). Formally,

$$\phi^* = \arg\min_{\phi} D(q_\phi(\boldsymbol{\theta})||p(\boldsymbol{\theta}|\boldsymbol{x}_o))$$

with $q_{\phi^*}(\boldsymbol{\theta}) = p(\boldsymbol{\theta}|\boldsymbol{x}_o) \iff D(q_{\phi^*}(\boldsymbol{\theta})||p(\boldsymbol{\theta}|\boldsymbol{x}_o)) = 0$. Recent work has introduced normalizing flows as a variational family for VI (Ranganath et al., 2014; Agrawal et al., 2020; Rezende & Mohamed, 2015). Normalizing flows define a distribution $q_\phi(\boldsymbol{\theta})$ by learning a bijection $T_\phi$ which transforms a simpler distribution into a complex distribution $p(\boldsymbol{\theta}|\boldsymbol{x}_o)$. Normalizing flows provide a highly flexible variational family, while at the same time allowing low variance gradient estimation of an expectation by the reparameterization trick, i.e. $\nabla_\phi\mathbb{E}_{\boldsymbol{\theta}\sim q_\phi}[f(\boldsymbol{\theta})] = \mathbb{E}_{\boldsymbol{\theta}_0\sim q_0}[\nabla_\phi f(T_\phi(\boldsymbol{\theta}_0))]$ with $\boldsymbol{\theta} = T_\phi(\boldsymbol{\theta}_0)$ (Kingma & Welling, 2014; Rezende et al., 2014; Rezende & Mohamed, 2015).

## 3 SEQUENTIAL NEURAL VARIATIONAL INFERENCE (SNVI)

### 3.1 KEY INGREDIENTS

We propose a framework to use variational inference (VI) for simulation-based inference. Our method consists of three parts: A learnable likelihood (or likelihood-ratio) model, a posterior model (typically parameterized as a normalizing flow) to be learned with VI, and sampling importance resampling (SIR) (Rubin, 1988) to refine the accuracy of the posterior (Fig. 1). The likelihood(-ratio) model $\ell_\psi(\boldsymbol{x}|\boldsymbol{\theta})$ learns to approximate the likelihood $p(\boldsymbol{x}|\boldsymbol{\theta})$ or the likelihood-ratio $\frac{p(\boldsymbol{x}|\boldsymbol{\theta})}{p(\boldsymbol{x})}$ from pairs of parameters and simulation outputs $(\boldsymbol{\theta},\boldsymbol{x})$. We use the term SNLVI to refer to SNVI with likelihoods, and SNRVI with likelihood-ratios. After a likelihood(-ratio) model has been trained, the posterior model $q_\phi(\boldsymbol{\theta})$ is trained with variational inference using normalizing flows. Finally, SIR is used to correct potential inaccuracies in the posterior $q_\phi(\boldsymbol{\theta})$– as we will show below, the SIR step leads to empirical improvements at modest computational overhead. To refine the likelihood(-ratio) model and the posterior, the procedure can be repeated across several 'rounds'. We opt to sample the parameters $\boldsymbol{\theta}$ from the previous posterior estimate $q_\phi(\boldsymbol{\theta})$, but other strategies for active learning (e.g. Lueckmann et al., 2019b) could be plugged into SNVI. The algorithm is summarized in Alg. 1. We will now describe three variational objectives that can be used with SNVI, the SIR procedure to refine the posterior, and a strategy for dealing with invalid simulation outputs.

### 3.2 VARIATIONAL OBJECTIVES FOR SBI

Because of the expressiveness of normalizing flows, the true posterior can likely be approximated well by a member of the variational family (Papamakarios et al., 2021). Thus, the quality of the

---

**Algorithm 1:** SNVI

---

1   **Inputs:** prior $p(\boldsymbol{\theta})$, observation $\boldsymbol{x}_o$, divergence $D$, simulations per round $N$, number of rounds $R$, selection strategy $\mathcal{S}$.

2   **Outputs:** Approximate likelihood $\ell_\psi$ and variational posterior $q_\phi$.

3   **Initialize:** Proposal $\tilde{p}(\boldsymbol{\theta}) = p(\boldsymbol{\theta})$, simulation dataset $\mathcal{X} = \{\}$

4   **for** $r \in [1, ..., R]$ **do**

5      **for** $i \in [1, ..., N]$ **do**

6         $\boldsymbol{\theta}_i = \mathcal{S}(\tilde{p}, \ell_\phi, p)$ ;                   `// sample` $\boldsymbol{\theta}_i \sim \tilde{p}(\boldsymbol{\theta})$

7         simulate $\boldsymbol{x}_i \sim p(\boldsymbol{x}|\boldsymbol{\theta}_i)$ ;          `// run the simulator on` $\boldsymbol{\theta}_i$

8         add $(\boldsymbol{\theta}_i, \boldsymbol{x}_i)$ to $\mathcal{X}$

9      **end**

10     (re-)train $\ell_\psi$;    $\psi^* = \arg\min_\psi -\frac{1}{N} \sum_{(\boldsymbol{x}_i, \boldsymbol{\theta}_i) \in \mathcal{X}} \log \ell_\psi(\boldsymbol{x}_i|\boldsymbol{\theta}_i)$ ; `// or SNRE loss`

11     (re-)train $q_\phi$;    $\phi^* = \arg\min_\phi D(q_\phi(\boldsymbol{\theta}) || p(\boldsymbol{\theta}|\boldsymbol{x}_o))$ with

$$p(\boldsymbol{\theta}|\boldsymbol{x}_o) \propto p(\boldsymbol{x}_o|\boldsymbol{\theta})p(\boldsymbol{\theta}) \approx \ell_{\psi^*}(\boldsymbol{x}_o|\boldsymbol{\theta})p(\boldsymbol{\theta})$$

12     $\tilde{p}(\boldsymbol{\theta}) = q_\phi(\boldsymbol{\theta})$

13   **end**

---

variational approximation is strongly linked to the ability to achieve the best possible approximation through optimization, which in turn depends on the choice of variational objective $D$. Using the reverse Kullback-Leibler Divergence (rKL) as proposed by Wiqvist et al. (2021) can give rise to mode-seeking behaviour and $q_\phi$ might not cover all regions of the posterior (Bishop, 2006; Blei et al., 2017). As a complementary approach, we suggest and evaluate three alternative variational objectives that induce a *mass-covering* behaviour and posit that this strategy will be particularly important in sequential schemes.

**1. Forward KL divergence (fKL)**   In contrast to the reverse KL (rKL), the forward Kullback-Leibler divergence (fKL) is mass-covering (Bishop, 2006). Wan et al. (2020) minimize the following upper bound to the evidence, which implicitly minimizes the fKL: $\mathcal{L}(\phi) = \mathbb{E}_{\boldsymbol{\theta} \sim q_\phi} [w(\boldsymbol{\theta}) \log (w(\boldsymbol{\theta}))]$ with $w(\boldsymbol{\theta}) = p(\boldsymbol{x}_o, \boldsymbol{\theta})/q_\phi(\boldsymbol{\theta})$. This expression is hard to estimate with samples: If $q_\phi(\boldsymbol{\theta})$ is different from $p(\boldsymbol{x}_o, \boldsymbol{\theta})$ then $w(\boldsymbol{\theta}) \approx 0$ for most $\boldsymbol{\theta} \sim q_\phi(\boldsymbol{\theta})$, thus $\nabla_\phi \mathcal{L}(\phi) \approx 0$, which would prevent learning (see Appendix Sec. A.3).

To overcome this problem, we rewrite the fKL using self-normalized importance sampling (Jerfel et al., 2021). Let $\boldsymbol{\theta}_1, \ldots, \boldsymbol{\theta}_N \sim \pi$ be samples from an arbitrary proposal distribution $\pi$. We then minimize the loss:

$$\mathcal{L}_{\text{fKL}}(\phi) = D_{KL}(p||q_\phi) \approx \sum_{i=1}^{N} \frac{w(\boldsymbol{\theta}_i)}{\sum_{j=1}^{N} w(\boldsymbol{\theta}_j)} \log \left( \frac{p(\boldsymbol{x}_o, \boldsymbol{\theta}_i)}{q_\phi(\boldsymbol{\theta}_i)} \right)$$

where $w(\boldsymbol{\theta}) = p(\boldsymbol{x}_o, \boldsymbol{\theta})/\pi(\boldsymbol{\theta})$. As a self-normalized importance sampling scheme, this estimate is biased, but the bias vanishes at rate $\mathcal{O}(1/N)$ (Hesterberg, 2003). In our experiments, we use $\pi = q_\phi$, which provides a good proposal when $q_\phi$ is close to $p$ (Chatterjee & Diaconis, 2018). Even though $q_\phi$ will differ from $p$ initially, sufficient gradient information is available to drive $q_\phi$ towards $p$, as we demonstrate in Appendix Sec. A.3.

**2. Importance weighted ELBO**   The importance weighted ELBO (IW-ELBO) introduced by Burda et al. (2016) uses the importance-weighted gradient of the evidence lower bound (ELBO). It minimizes the KL divergence between the self-normalized importance sampling distribution of $q_\phi$ and the posterior and thus provides a good proposal for sampling importance resampling (Cremer et al., 2017; Domke & Sheldon, 2018; Ranganath et al., 2014). It can be formulated as

$$\mathcal{L}_{IW}^{(K)}(\phi) = \mathbb{E}_{\boldsymbol{\theta}_1, \ldots, \boldsymbol{\theta}_k \sim q_\phi} \left[ \log \frac{1}{K} \sum_{k=1}^{K} \frac{p(\boldsymbol{x}_o, \boldsymbol{\theta}_k)}{q_\phi(\boldsymbol{\theta}_k)} \right].$$

To avoid a low SNR of the gradient estimator (Rainforth et al., 2018), we use the 'Sticking the Landing' (STL) estimator introduced by Roeder et al. (2017).

**3. Rényi $\alpha$-divergences** Rényi $\alpha$-divergences are a divergence family with a hyperparameter $\alpha$ which allows to tune the mass-covering (or mode-seeking) behaviour of the algorithm. For $\alpha \to 1$, the divergence approaches the rKL. For $\alpha < 1$, the divergence becomes more mass-covering, for $\alpha > 1$ more mode-seeking. We use $\alpha = 0.1$ in our experiments. A Rényi variational bound was established by Li & Turner (2016) and is given by

$$\mathcal{L}_\alpha(\phi) = \frac{1}{1-\alpha} \log \left( \mathbb{E}_{\boldsymbol{\theta} \sim q_\phi} \left[ \left( \frac{p(\boldsymbol{x}_o, \boldsymbol{\theta})}{q_\phi(\boldsymbol{\theta})} \right)^{1-\alpha} \right] \right)$$

For $\alpha = 0$, $\mathcal{L}_\alpha$ is a single sample Monte Carlo estimate of the IW-ELBO (when using $K$ samples to estimate the expectation in $\mathcal{L}_\alpha(\phi)$) and thus also suffers from a low SNR as $\alpha \to 0$ (Rainforth et al., 2018; Li & Turner, 2016). Just as for the IW-ELBO, we alleviate this issue by combining the $\alpha$-divergences with the STL estimator.

### 3.3 Sampling Importance Resampling

After the variational posterior has been trained, $q_\phi$ approximates $\ell_\psi(\boldsymbol{x}_o|\boldsymbol{\theta})p(\boldsymbol{\theta})/Z$ with normalization constant $Z$. We propose to improve the quality of posterior samples by applying Sampling Importance Resampling (SIR) (Rubin, 1988). We sample $K = 32$ samples from $\boldsymbol{\theta} \sim q_\phi(\boldsymbol{\theta})$, compute the corresponding importance weights $w_i = \ell_\psi(\boldsymbol{x}_o|\boldsymbol{\theta}_i)p(\boldsymbol{\theta}_i)/q_\phi(\boldsymbol{\theta}_i)$ and resample a single sample from a categorical distribution whose probabilities equal the normalized importance weights (details in Appendix Sec. A.4). This strategy enriches the variational family with minimal computational cost (Agrawal et al., 2020). SIR is particularly useful when $q_\phi(\boldsymbol{\theta})$ covers the true posterior and is thus well-suited for the objectives described above (see Appendix Fig. 6).

### 3.4 Excluding invalid data

Simulators may produce unreasonable or undefined values (e.g. NaN), as we will also see in the pyloric network model described later. In posterior estimation methods (SNPE), one can simply remove these 'invalid' simulations from the training dataset, and the trained neural density estimator will still approximate the true posterior (Lueckmann et al., 2017). However, as we show in Appendix Sec. A.6, this is not the case for likelihood(-ratio)-methods– when 'invalid' simulations are removed, the network will converge to $\ell_\psi(\boldsymbol{x}_o|\boldsymbol{\theta}) \approx \frac{1}{Z}p(\boldsymbol{x}_o|\boldsymbol{\theta})/p(\text{valid}|\boldsymbol{\theta})$, i.e. the learned likelihood-function will be biased towards parameter regions which often produce 'invalid' simulations. This prohibits any method that estimates the likelihood(-ratio) (i.e. SNVI, SNLE, SNRE) from excluding 'invalid' simulations, and would therefore prohibit their use on models that produce such data.

To overcome this limitation of likelihood-targeting techniques, we propose to estimate the bias-term $p(\text{valid}|\boldsymbol{\theta})$ with an additional feedforward neural network $c_\zeta(\boldsymbol{\theta}) \approx p(\text{valid}|\boldsymbol{\theta})$ (details in Appendix Sec. A.6). Once trained, $c_\zeta(\boldsymbol{\theta})$ can be used to correct for the bias in the likelihood network. Given the (biased) likelihood network $\ell_\psi(\boldsymbol{x}_o|\boldsymbol{\theta})$ and the correction factor $c_\zeta(\boldsymbol{\theta})$, the posterior distribution is proportional to

$$\mathcal{P}(\boldsymbol{\theta}) = \ell_\psi(\boldsymbol{x}_o|\boldsymbol{\theta})p(\boldsymbol{\theta})c_\zeta(\boldsymbol{\theta}) \propto p(\boldsymbol{x}_o|\boldsymbol{\theta})p(\boldsymbol{\theta}) \propto p(\boldsymbol{\theta}|\boldsymbol{x}_o).$$

We sample from this distribution with VI in combination with SIR. Details, proof and extension to SNRVI in Appendix Sec. A.6. The additional network $c_\zeta(\boldsymbol{\theta})$ is only required in models which can produce invalid simulations. This is not the case for the toy models in Sec. 4.2, but it is required in the model in Sec. 4.3. Alg. 1 shows SNVI without the additional bias-correction step, Appendix Alg. 3 shows the method with correction.

## 4 Experiments

We demonstrate the accuracy and the computational efficiency of SNVI on several examples. First, we apply SNVI to an illustrative example to demonstrate its ability to capture complex posteriors without mode-collapse. Second, we compare SNVI to alternative methods on several benchmark tasks. Third, we demonstrate that SNVI can obtain the posterior distribution in models with many parameters by applying it to a neuroscience model of the pyloric network in the crab *Cancer borealis*.

Figure 2: **A** Posterior approximations of SNLE, SNPLA, and SNVI+fKL for the two moons benchmark example. **B** Runtime of all algorithms.

## 4.1 Illustrative example: Two moons

We use the 'two moons' simulator (Greenberg et al., 2019) to illustrate the ability of SNVI to capture complex posterior distributions. The two moons simulator has two parameters with a uniform prior and generates a posterior that has both local and global structure. Fig. 2A shows the ground truth posterior distribution as well as approximations learned by several methods using $10^5$ simulations.

SNLE with MCMC (in the form of Slice Sampling with axis-aligned updates (Neal, 2003)) can recover the bimodality when running 100 chains in parallel (Lueckmann et al., 2021) (not shown: individual chains typically only explore a single mode). SNPLA, which is based on the mode-seeking rKL (and could thus also be considered as SNVI+rKL, see Appendix Sec. A.7) captures only a single mode. In contrast, SNLVI (using the fKL and SIR, denoted as SNVI+fKL) recovers both the local and the global structure of the posterior accurately. In terms of runtime, SNPLA and SNVI+fKL are up to twenty times faster than 100 chain MCMC in our implementation (Fig. 2B), and two to four orders of magnitude faster than single chain MCMC (single chain not shown, the relative speed-up for multi-chain MCMC is due to vectorization).

## 4.2 Results on benchmark problems

We compare the accuracy and computational cost of SNVI to that of previous methods, using SBI benchmark tasks (Lueckmann et al., 2021):

**Bernoulli GLM:** Generalized linear model with Bernoulli observations. Inference is performed on 10-dimensional sufficient summary statistics of the originally 100 dimensional raw data. The resulting posterior is 10-dimensional, unimodal, and concave.

**Lotka Volterra:** A traditional model in ecology (Wangersky, 1978), which describes a predator-prey interaction between species, illustrating a task with complex likelihood and unimodal posterior.

**Two moons:** Same as described in the previous section.

**SLCP:** A task introduced by Papamakarios et al. (2019) with a simple likelihood and complex posterior. The prior is uniform, the likelihood has Gaussian noise but is nonlinearly related to the parameters, resulting in a posterior with four symmetrical modes.

For each task, we perform inference for ten different runs, each with a different observation. As performance metric, we used classifier 2-sample tests (C2ST) (best is 0.5, worst is 1.0) (Friedman, 2004; Lopez-Paz & Oquab, 2017). For each method, we perform inference given a total of $10^3$, $10^4$ and $10^5$ simulations, evenly distributed across ten rounds of simulation and training. Details on the hyperparameters are provided in Appendix Sec. A.8, details on results in Appendix Fig. 9, comparisons to the forward KL without self-normalized weights as well as to the IW-ELBO and the $\alpha$-divergences without STL in Appendix Fig. 11.

We show results for two reference methods, SNLE with MCMC sampling, and SNPLA, and compare them to three variants of SNLVI using the forward KL (SNVI+fKL), the importance-weighted ELBO (SNVI+IW) as well as an alpha-divergence (SNVI+$\alpha$). We find that all three SNVI-variants achieve performance comparable to MCMC across all four tasks (Fig. 3 A-D, left), and outperform SNPLA on the two tasks with multi-modal posteriors (Two moons and SLCP). We find that omitting the SIR-adjustment (dotted lines) leads to a small but consistent degradation in inference

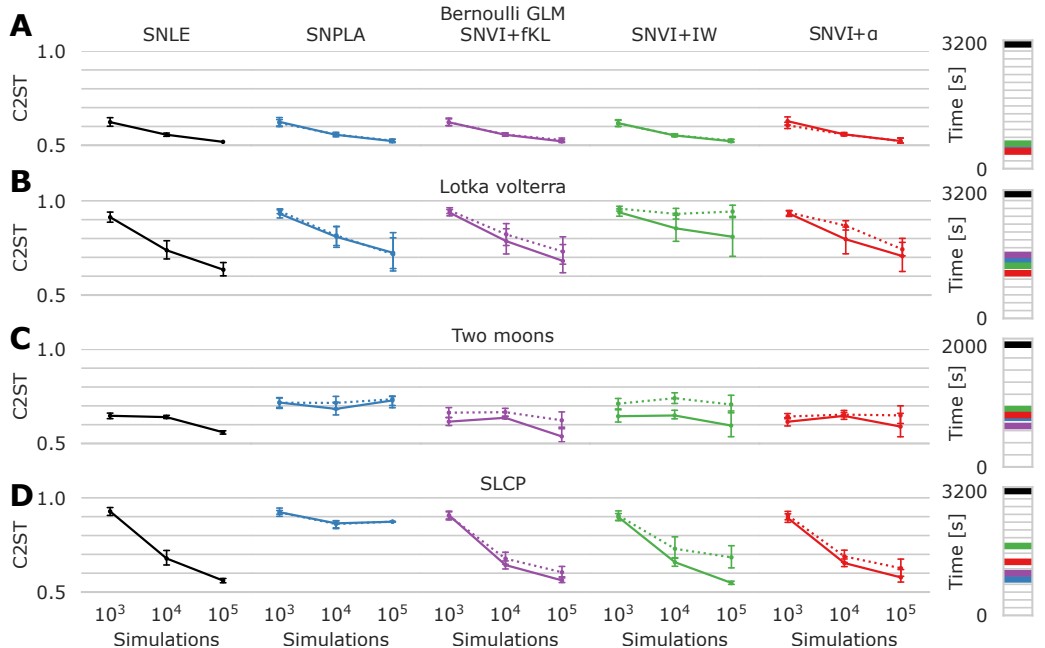

Figure 3: C2ST benchmark results for SNVI with likelihood-estimation (SNLVI) for four models, Bernoulli GLM (**A**), Lotka volterra (**B**), Two moons (**C**) and SLCP (**D**). Each point represents the average metric value for ten different observations, as well as the confidence intervals. Bars on the right indicate the average runtime. Two reference methods: SNLE with MCMC sampling, and SNPLA (which uses rKL), as well as three variants of SNVI, with forward KL (SNVI+fKL), importance-weighted ELBO (SNVI+IW) and $\alpha$-divergence (SNVI+$\alpha$). Dotted lines: Performance when not using SIR.

performance for all SNVI-variants, but not for SNPLA with the rKL: When using the rKL, the approximate posterior $q_\phi$ is generally narrower than the posterior and thus ill-suited for SIR (Appendix Fig. 6). Qualitatively similar results were found when using likelihood-ratio approaches with the same hyperparameters, see Appendix Fig. 10.

In terms of runtime, all three variants of SNLVI are substantially faster than SNLE on every task (bars in Fig. 3 on the right), in some cases by more than an order of magnitude. When using likelihood-ratio estimation, MCMC with 100 chains can be as fast as SNRVI on tasks with few parameters (Appendix Fig. 10). On tasks with many parameters, however, SNRVI is significantly faster than SNRE (see e.g. Bernoulli GLM with 10 parameters).

### 4.3 INFERENCE IN A NEUROSCIENCE MODEL OF THE PYLORIC NETWORK

Finally, we applied SNVI to a simulator of the pyloric network in the stomatogastric ganglion (STG) of the crab *Cancer Borealis*, a well-characterized circuit producing rhythmic activity. The model consists of three model neurons (each with eight membrane conductances) with seven synapses (31 parameters in total) and produces voltage traces that can be characterized with 15 established summary statistics (Prinz et al., 2003; 2004). In this model, disparate parameter sets can produce similar activity, leading to a posterior distribution with broad marginals but narrow conditionals (Prinz et al., 2004; Gonçalves et al., 2020). Previous work has used millions of simulations from prior samples and performed amortized inference with NPE (18 million simulations in Gonçalves et al. (2020), 9 million in Deistler et al. (2021)). Sequential neural posterior estimation (SNPE) struggles on this problem due to leakage, whereas SNLE and SNRE with MCMC are inefficient (Durkan et al., 2020). Here, we apply SNVI to identify the posterior distribution given an extracellular recording of the stomatogastric motor neuron (Fig. 4A) (Haddad & Marder, 2021; 2018). We demonstrate that SNVI can perform multi-round inference and obtains the posterior distribution with only 350,000 simulations – 25 times fewer than previous methods!

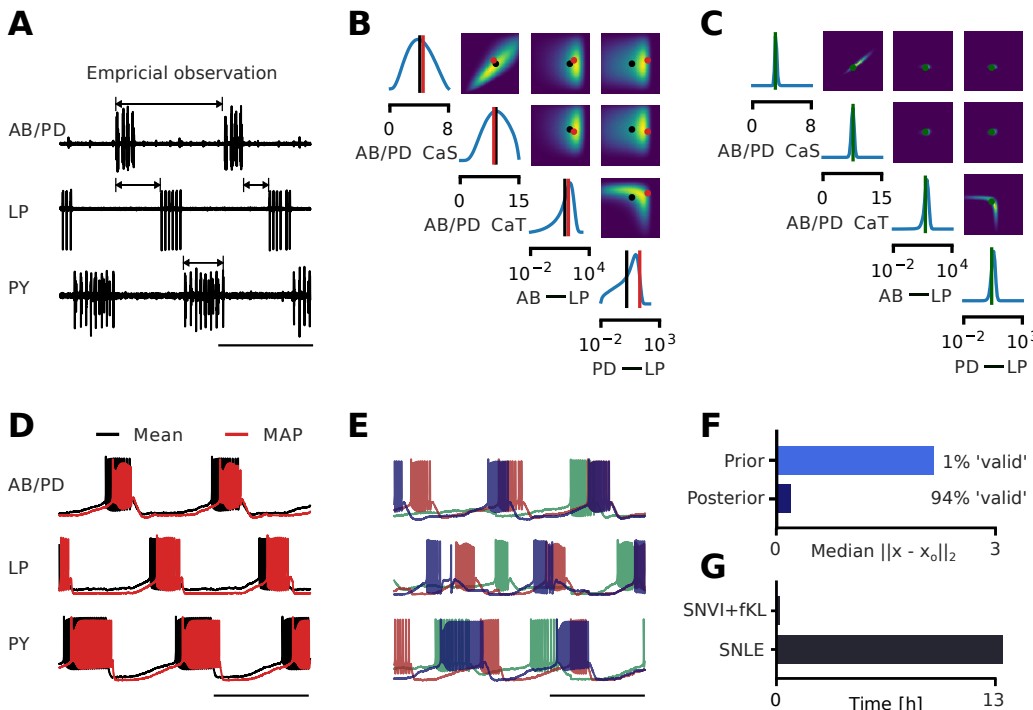

Figure 4: **(A)** Empirical observation, arrows indicate some of the summary statistics. Scale bar is one second. **(B)** Cornerplot showing a subset of the marginal and pairwise marginal distributions of the 31-dimensional posterior (full posterior in Appendix Fig. 12). Red dot: MAP. Black dot: Posterior mean. **(C)** Conditional distributions $p(\boldsymbol{\theta}_{i,j}|\boldsymbol{x}, \boldsymbol{\theta}_{\neq i,j})$. Green dot shows the sample on which we condition. **(D)** Simulated traces from the posterior mean and MAP. Scale bar is one second. **(E)** Simulated traces of three posterior samples. **(F)** Posterior predictive and prior predictive median (z-scored) distances from the observation. **(G)** Time required to obtain 10k samples: SNVI takes 11 minutes and SNLE with 100-chain MCMC 808 minutes, i.e. over 13 hours.

We ran SNVI with a likelihood-estimator with the fKL divergence (SNVI+fKL) and SIR. Since the simulator produces many invalid summary statistics (e.g. gaps between bursts cannot be defined if there are no bursts) we employed the strategy described in Sec. 3.4. Because only $1\%$ of the simulations from prior samples are valid (Fig. 4F), we used 50,000 simulations in the first round and continued for 30 rounds with 10,000 simulations each.

The posterior is complex and reveals strong correlations and nonlinear relationships between parameters (Fig. 4B showing 4 out of 31 dimensions, full posterior in Appendix Fig. 12). The conditional distributions $p(\boldsymbol{\theta}_{i,j}|\boldsymbol{x}, \boldsymbol{\theta}_{\neq i,j})$ given a posterior sample (Fig. 4C) are narrow, demonstrating that parameters have to be finely tuned to generate the summary statistics of the experimentally measured activity. We used posterior predictive checks to inspect the quality of the posterior. When simulating data from the posterior mean and posterior mode (MAP), we find that both of them match the statistics of the experimental activity (Fig. 4D). Similarly, samples from the posterior distribution closely match statistics of the experimental activity (Fig. 4E). Out of 10,000 posterior samples, 9366 ($\approx 94\%$) generated activity with well-defined summary statistics (compared to $1\%$ of prior samples). For the samples which generate well-defined summary statistics, the (z-scored) median distance between the observed data $\boldsymbol{x}_o$ and generated activity is smaller for posterior samples than for prior samples (Fig. 4F). We emphasize that an application of SNLE with MCMC would be estimated to take an additional $400$ hours, due to 30 rounds of slow MCMC sampling (Fig. 4G) that would be required– instead of 27 hours for SNVI. Likewise, when running SNPE-C on this example, only one out of 2 million samples was within the prior bounds after the second round, requiring computationally expensive rejection sampling (Greenberg et al., 2019; Durkan et al., 2020). Finally, we note that the additional neural network $c_\zeta(\boldsymbol{\theta})$ (required to correct for the effect of invalid simulations) can be learned robustly and with low computational cost (see Appendix Fig. 13 for runtime).

These results show that SNVI makes it possible to overcome the limitations of previous methods and allows sequential neural simulation-based inference methods to effectively and robustly scale to challenging inference problems of scientific interest. While it is difficult to rigorously evaluate the accuracy of the obtained posterior distribution due to a lack of ground truth, we observed that almost all posterior predictives have well-defined summary statistics (94% vs 80% in Gonçalves et al. (2020)) and that the posterior predictives closely match $x_o$.

## 5 DISCUSSION

We introduced Sequential Neural Variational Inference (SNVI), an efficient, flexible, and robust approach to perform Bayesian inference in models with an intractable likelihood. We achieve this by combining likelihood-estimation (or likelihood-ratio estimation) with variational inference, further improved by using SIR for refining posteriors. We demonstrate that SNVI reduces the computational cost of inference while maintaining accuracy. We applied our approach to a neuroscience model of the pyloric network with 31 parameters and showed that it is 25 times more efficient than previous methods. Our results demonstrate that SNVI is a scalable and robust method for simulation-based inference, opening up new possibilities for Bayesian inference in models with intractable likelihoods.

We selected three variational objectives for SNVI which induce mass-covering behaviour and are, therefore, well suited as a proposal for sampling from complex posterior distributions. We empirically evaluated all of these methods in terms of runtime and accuracy on four benchmark tasks. We found that, while their performance differed when using the raw VI output, they all showed similar performance after an additional, computationally cheap, sampling importance resampling (SIR) step. After the SIR step, all methods had similar accuracy as MCMC, and all methods outperformed a mode-seeking variational objective (reverse KL) which was used in a previously proposed method Wiqvist et al. (2021). Our results suggest that mass-covering VI objectives (regardless of their exact implementation) provide a means to perform fast and accurate inference in models with intractable likelihood, without loss of accuracy compared to MCMC. In Appendix Sec. A.2, we provide technical recommendations for choosing a variational objective for specific problems.

A common approach in sequential methods is to use the current posterior estimate as the proposal distribution for the next round, but more elaborate active-learning strategies for choosing new simulations are possible (Papamakarios & Murray, 2016; Papamakarios et al., 2019; Lueckmann et al., 2019a). SNVI can flexibly be combined with any active learning scheme, and unlike neural likelihood(-ratio) methods, does not require expensive MCMC sampling for updating posterior estimates. While this comes at the cost of having to train two neural networks (a likelihood-model and a posterior-model), the cost of training these neural networks is often negligible compared to the cost of simulations. Another method that trains both a likelihood- and a posterior network is Posterior-Aided Regularization (Kim et al., 2021), which regularizes the likelihood-estimate with a simultaneously trained posterior-estimate. This improves the modelling of multimodal posteriors, but the method still requires MCMC and thus scales poorly with the number of samples and dimensions. Likelihood-free variational inference (Tran et al., 2017) avoids learning a likelihood model by learning an implicit posterior distribution, but it requires an adversarial training objective which can be difficult to optimize and requires extensive hyperparameter tuning (Huszár, 2017). Ong et al. (2018) is another method that performs variational inference with a synthetic likelihood, but their approach requires that the summary statistics are approximately Gaussian in order to obtain unbiased estimates of the log-likelihood.

Overall, SNVI combines the desirable properties of current methods: It can be combined with any active learning scheme, it can flexibly combine information from multiple datapoints, it returns a posterior distribution that can be sampled quickly, and it can robustly deal with missing data. SNVI speeds up inference relative to MCMC-based methods, sometimes by orders of magnitude, and can perform inference in large models with many parameters. SNVI therefore has potential to provide a new 'go-to' approach for simulation-based inference, and to open up new application domains for simulation-based Bayesian inference.

## 6 REPRODUCIBILITY STATEMENT

We used the configuration manager hydra to track the configuration and seeds of each run (Yadan, 2019). The results shown in this paper can be reproduced with the git repository `https://github.com/mackelab/snvi_repo`. The algorithms developed in this work are also available in the sbi toolbox (Tejero-Cantero et al., 2020). All simulations and runs were performed on a high-performance computer. For each run, we used 16 CPU cores (Intel family 6, model 61) and 8GB RAM.

## 7 ACKNOWLEDGEMENTS

We thank Jan-Matthis Lueckmann for insightful comments on the manuscript. This work was funded by the German Research Foundation (DFG; Germany's Excellence Strategy MLCoE – EXC number 2064/1 PN 390727645) and the German Federal Ministry of Education and Research (BMBF; Tübingen AI Center, FKZ: 01IS18039A).

## 8 ETHICS STATEMENT

We used data recorded from animal experiments in the crab *Cancer borealis*. The data we used were recorded for a different, independent study and have recently been made publicly available (Haddad & Marder, 2018; 2021). While simulation-based inference has the potential to greatly accelerate scientific discovery across a broad range of disciplines, one could also imagine undesired use-cases of SBI.

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

# A APPENDIX

## A.1 GRADIENTS OF THE DIVERGENCES

For completeness, we provide the gradients of the divergences introduced in Sec. 3.2.

**Forward Kullback-Leibler divergence** The gradient estimate is given by

$$\nabla_\phi \mathcal{L}_{\text{fKL}}(\phi) = -\mathbb{E}_{\boldsymbol{\theta} \sim p} \left[ \nabla_\phi \log \left( q_\phi(\boldsymbol{\theta}) \right) \right] \approx - \sum_{i=1}^{N} \frac{w(\boldsymbol{\theta}_i)}{\sum_{j=1}^{N} w(\boldsymbol{\theta}_j)} \nabla_\phi \log q_\phi(\boldsymbol{\theta}_i)$$

**Importance-weighted ELBO** A gradient estimator is given by

$$\nabla_\phi \mathcal{L}_{IW}^{(K)}(\phi) = \mathbb{E}_{\boldsymbol{\theta}_1, \dots, \boldsymbol{\theta}_K \sim q_\phi} \left[ \sum_{i=1}^{K} \tilde{w}(\boldsymbol{\theta}_i) \nabla_\phi \log \left( \frac{p(\boldsymbol{x}_o, \boldsymbol{\theta}_i)}{q_\phi(\boldsymbol{\theta}_i)} \right) \right] \quad \tilde{w}(\boldsymbol{\theta}_i) = \frac{w(\boldsymbol{\theta}_i)}{\sum_{i=1}^{K} w(\boldsymbol{\theta}_i)}$$

where $w(\boldsymbol{\theta}) = p(\boldsymbol{x}_o, \boldsymbol{\theta})/q_\phi(\boldsymbol{\theta})$.

**Rényi $\alpha$-divergence** A biased gradient estimator using the reparameterization trick can be written as:

$$\nabla_\phi \mathcal{L}_\alpha(\phi) = \mathbb{E}_{\boldsymbol{\theta} \sim q_\phi} \left[ \tilde{w}_\alpha(\boldsymbol{\theta}) \nabla_\phi \log \left( \frac{p(\boldsymbol{x}_o, \boldsymbol{\theta})}{q_\phi(\boldsymbol{\theta})} \right) \right] \quad \tilde{w}_\alpha = \frac{w(\boldsymbol{\theta})^{1-\alpha}}{\mathbb{E}_{\boldsymbol{\theta} \sim q_\phi} [w(\boldsymbol{\theta})^{1-\alpha}]},$$

where $w(\boldsymbol{\theta}) = p(\boldsymbol{x}_o, \boldsymbol{\theta})/q_\phi(\boldsymbol{\theta})$. This gradient estimator is biased towards the rKL but the bias vanishes as more samples are used for the Monte Carlo approximation (Li & Turner, 2016).

## A.2 CHOICE OF DIVERGENCE

In Fig. 4.2, we demonstrated that all mass-covering objectives perform similarly in terms of accuracy and runtime on the problems we considered. We here give technical recommendations for choosing a variational objective:

  (i) *Closed-form posterior:* The variational posterior provides a closed-form approximation to the posterior, but this is no longer the case when SIR is used. While, in our results, all three approaches performed similarly *with* SIR, they can differ in their performance *without it*, and the forward KL and the $\alpha$-divergence provided better approximations than the IW-ELBO. Thus, if one seeks a posterior density that can be evaluated in closed-form, our results suggest to use the forward KL or the $\alpha$-divergence.

 (ii) *Dimensionality of the parameter space*: We use an autoregressive normalizing flow as variational family. These flows are very expressive, yet the computation time of their forward or backward passes scale with the dimensionality of $\boldsymbol{\theta}$ (Papamakarios et al., 2017; Kingma et al., 2016; Durkan et al., 2019a). The IW-ELBO and the $\alpha$-divergences only require forward passes, whereas the forward KL requires forward and backward passes, thus making the forward KL expensive for high-dimensional parameter spaces. The STL estimator used in the IW-ELBO and the $\alpha$-divergences also requires forward and backward passes. We found that the STL estimator improves performance of the IW-ELBO only weakly (Fig. 11). Thus, in cases where computational cost is critical, our results suggest that using the IW-ELBO without the STL can give high accuracy at low computational cost. Another way to reduce computational cost is to use alternative architectures for the normalizing flow, e.g. coupling layers (Durkan et al., 2019a; Papamakarios et al., 2021).

(iii) *Trading-off the mass-covering property with computational cost*: For $\alpha$-divergences, one can trade-off the extent to which the divergence is mass-covering by choosing the value of $\alpha$ (low $\alpha$ is more mass-covering). As shown in Fig. 11, high values of $\alpha$ benefit less from using the STL estimator. Thus, in cases where mass-covering behavior of the algorithm is less crucial, the STL estimator can be waived, leading to lower computational cost because the normalizing flow requires only forward passes (see point (ii)).

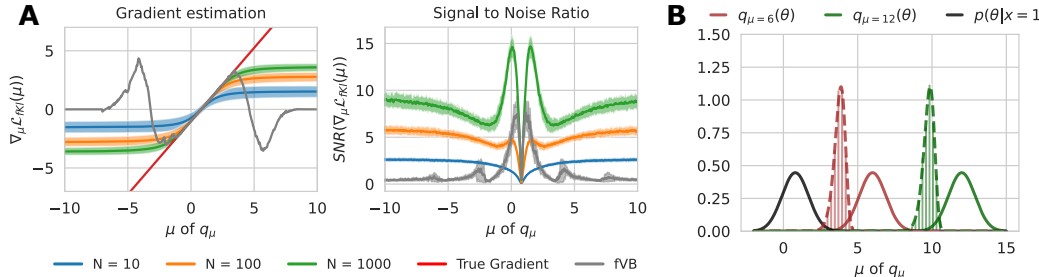

Figure 5: **A** Left: Gradient estimation on the Gaussian example. For values of $\mu$ around $\mu^* = 4/5$, all estimators provide good gradients. As $\mu$ is farther from $\mu^*$, the forward variational bound (fVB) (grey) vanishes, whereas the self-normalized fVB approaches a constant. Right: SNR for the fVB and the self-normalized fVB. **B** Theoretical and empirical densities $p_R(r)$ for $\mu = 6$ and $\mu = 12$.

### A.3 OVERCOMING VANISHING GRADIENTS IN THE FORWARD KL ESTIMATOR

We use an estimator of the forward Kullback-Leibler divergence (fKL) that is based on self-normalized importance sampling. In this section, we demonstrate that this estimator moves the variational distribution $q_\phi(\boldsymbol{\theta})$ towards the target density $p(\boldsymbol{x}_o, \boldsymbol{\theta})$ even if $q_\phi(\boldsymbol{\theta})$ and $p(\boldsymbol{x}_o, \boldsymbol{\theta})$ differ strongly.

For this analysis, we consider a Gaussian toy example with prior $p(\boldsymbol{\theta}) = \mathcal{N}(\boldsymbol{\theta}; 0, 4)$, likelihood $p(\boldsymbol{x}|\boldsymbol{\theta}) = \mathcal{N}(\boldsymbol{x}; \boldsymbol{\theta}, 1)$, and observation $\boldsymbol{x}_o = 1$. The posterior distribution can be computed in closed-form as $p(\boldsymbol{\theta}|\boldsymbol{x}_o) = \mathcal{N}(\boldsymbol{\theta}; 4/5, 4/5)$. We aim to learn the posterior distribution using variational inference with the variational family $q_\mu(\boldsymbol{\theta}) = \mathcal{N}(\mu, 4/5)$ (note that $\mu$ is the only parameter). The best approximation within this family is $\mu^* = 4/5$.

We use this toy example to compare the gradient and the signal-to-noise ratio (SNR) of the self-normalized fKL estimator to the fKL estimator introduced by Wan et al. (2020). Fig. 5A (left) shows the gradient of the loss for different values of $\mu$. When $\mu \approx \mu^* = 4/5$, the fKL (gray) without self-normalization closely matches the true gradient (red). However, as $\mu$ is further from $\mu^*$, the fKL first points in the wrong direction and then vanishes, which prevents learning. The self-normalized fKL (blue, orange, green) closely matches the gradient around $\mu \approx \mu^* = 4/5$ and does not vanish for $\mu$ that are far from $\mu^*$. The gradient is stronger if more samples $N$ are used to approximate the fKL. Similarly, the $\text{SNR}(\nabla_\phi \mathcal{L}(\phi)) = |\mathbb{E}[\nabla_\phi \mathcal{L}(\phi)]/\sqrt{\text{Var}(\nabla_\phi \mathcal{L}(\phi))}|$ does not vanish for $\mu$ that are far from $\mu^*$ for the self-normalized fKL.

To understand this behaviour of the self-normalized fKL, we computed an approximation to the gradient $\nabla_\mu \mathcal{L}_{\text{fKL}}(\mu)$ in this toy example. The fKL loss is given as:

$$\nabla_\mu \mathcal{L}_{\text{fKL}}(\mu) = -\mathbb{E}_{\boldsymbol{\theta} \sim q_\phi}\left[ \frac{w(\boldsymbol{\theta})}{\sum_{i=1}^{N} w(\boldsymbol{\theta})} \nabla_\mu \log q_\mu(\boldsymbol{\theta}) \right] \approx -\sum_{i=1}^{N} \frac{w(\boldsymbol{\theta}_i)}{\sum_{i=1}^{N} w(\boldsymbol{\theta}_i)} \nabla_\mu \log q_\mu(\boldsymbol{\theta}_i)$$

with weights $w(\boldsymbol{\theta}_i) = \frac{p(\boldsymbol{x}_o, \boldsymbol{\theta})}{q_\phi(\boldsymbol{\theta})}$. In the case where $q_\phi(\boldsymbol{\theta})$ differs strongly from $p(\boldsymbol{x}_o, \boldsymbol{\theta})$, the weights are often degenerate, i.e. the strongest weight is much larger than all others. In the worst case, $\tilde{w}(\boldsymbol{\theta}_i) = \frac{w(\boldsymbol{\theta}_i)}{\sum_{i=1}^{N} w(\boldsymbol{\theta}_i)} = 1$ for some $i$ and the gradient estimator reduces to

$$\nabla_\mu \mathcal{L}_{\text{fKL}}(\mu) = -\nabla_\mu \log q_\mu(\arg\max_{\boldsymbol{\theta}_1, \ldots, \boldsymbol{\theta}_N} w(\boldsymbol{\theta})) = -\nabla_\mu \log q_\mu(r)$$

The gradient of $\mu$ is thus determined by $r = \arg\max_{\boldsymbol{\theta}_1, \ldots, \boldsymbol{\theta}_N} w(\boldsymbol{\theta})$, which itself can be considered a draw from a random variable $R$. We will now derive the probability density $p_R(r)$ of $R$.

---

**Algorithm 2:** SIR

---

1 **Input**: $K$ the number of importance samples, proposal $q_\phi$, joint density
   $p(\boldsymbol{x}_o, \boldsymbol{\theta}) = \ell_\psi(\boldsymbol{x}_o|\boldsymbol{\theta})p(\boldsymbol{\theta})$
2 **for** $i \in [1, ..., K]$ **do**
3     $\boldsymbol{\theta}_i \sim q_\phi(\boldsymbol{\theta})$
4     $w_i = \frac{p(\boldsymbol{x}_o, \boldsymbol{\theta}_i)}{q_\phi(\boldsymbol{\theta}_i)}$
5 **end**
6 Each $\tilde{w}_i = w_i / \sum_{k=1}^{K} w_k$
7 $j \sim \text{Categorical}(\tilde{w})$
8 **return** $\boldsymbol{\theta}_j$

---

If $\mu > \mu^*$, then $w(\boldsymbol{\theta})$ is monotonically decreasing in $\boldsymbol{\theta}$ because $w(\boldsymbol{\theta}) \propto \frac{\mathcal{N}(\boldsymbol{\theta}; \mu^*, 4/5)}{\mathcal{N}(\boldsymbol{\theta}; \mu, 4/5)} \propto \exp(5/4 \cdot \boldsymbol{\theta}(\mu^* - \mu))$. The cumulative distribution function $F_R(R \leq r)$ can then be written as

$$F_R(R \leq r) = P(\arg\max_{\boldsymbol{\theta}_1, \dots, \boldsymbol{\theta}_n} w(\boldsymbol{\theta}) \leq r) = P(\min(\boldsymbol{\theta}_1, \dots, \boldsymbol{\theta}_n) \leq r)$$

$$= 1 - P(\min(\boldsymbol{\theta}_1, \dots, \boldsymbol{\theta}_n) > r) = 1 - \prod_{i=1}^{N} P(\boldsymbol{\theta}_i > r)$$

$$= 1 - (1 - F_{q_\phi}(r))^N$$

Thus, $R$ has the density $p_R(r) = \frac{d}{dr} F(R \leq r) = N(1 - F_{q_\phi}(r))^{N-1} q_\mu(r)$. The derivation is analogous for the case $\mu < \mu^*$. Because $\nabla_\mu \mathcal{L}_{\text{fKL}}(\mu) = \nabla_\mu \log q_\mu(r)$ for $r \sim p_R$, this allows us to compute the distribution of the gradient of $\mu$ (under the assumption that weights are degenerate).

We empirically validate this result on the Gaussian toy example. Fig. 5B (left) shows the true posterior distribution (black), the variational density $q_\mu(\boldsymbol{\theta})$ for $\mu = 6$ and $\mu = 10$ and the corresponding $p_R(r)$ for $N = 1000$. The theoretically computed density $p_R(r)$ (dashed lines) matches the empirically observed distribution of $\arg\max_{\boldsymbol{\theta}_i = 1 \dots N} w(\boldsymbol{\theta}_i)$. For almost every value of $r \sim p_R(r)$, the gradient $\nabla_\mu \log q_\mu(r)$ is negative, thus driving $\nabla_\mu \mathcal{L}_{\text{fKL}}(\mu)$ into the correct direction. For larger $N$, the distribution $p_R(r)$ shifts towards the true posterior distribution and thus also the gradient signal increases.

Notably, for $\mu$ that are even further from $\mu^*$, $\nabla_\mu \log q_\mu(r)$ remains relatively constant (Fig. 5B, right). This explains why the gradient $\nabla_\mu \mathcal{L}_{\text{fKL}}(\mu)$ becomes constant in Fig. 5A (left).

## A.4   IMPROVEMENT THROUGH SIR

We use Sampling Importance Resampling (SIR) to refine samples obtained from the variational posterior. The SIR procedure is detailed in Alg. 2 for drawing a single sample from the posterior. Consistent with Agrawal et al. (2020), we found that using SIR always helps to improve the approximation quality even when using complex variational families such as normalizing flows (compare dotted and solid lines in Fig. 3).

We visualize the benefits of SIR in Fig. 6 on an example which uses a Gaussian proposal distribution (i.e. variational family). SIR enhances the variational family and allows to approximate the bimodal target distribution. SIR particularly improves the posterior estimate when the proposal (i.e. the variational posterior) is overdispersed. This provides an explanation for why SIR is particularly useful for the mass-covering divergences used in SNVI, and less so for mode-covering divergences (as used in SNPLA).

## A.5   QUALITY OF THE LIKELIHOOD ESTIMATOR

The estimated variational posterior is based on the estimated likelihood $\ell(\boldsymbol{x}_o|\boldsymbol{\theta})$. It is thus important to accurately learn the likelihood from simulations. To be able to learn skewed or bimodal likelihoods, we use a conditional autoregressive normalizing flow for $\ell(\boldsymbol{x}|\boldsymbol{\theta})$ (Papamakarios et al., 2017; Kingma et al., 2016; Durkan et al., 2019a). Fig. 7 demonstrates that these flows can learn complex likelihoods (Papamakarios et al., 2019).

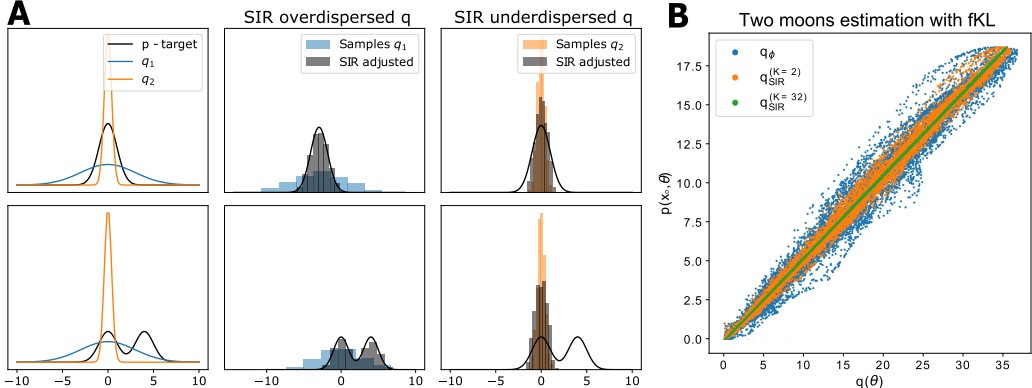

Figure 6: Visualization of SIR. **A** Toy example with a Gaussian proposal density. Left: Two toy examples with a Gaussian (top) or bimodal (bottom) target density. SIR (with $K = 32$) can extend the Gaussian density and refine the approximation if the proposal is overdispersed (middle), but helps less when it is too narrow (right). **B** SIR improvements on two moons example. We plot the joint density as learned by the likelihood-model $p(\boldsymbol{x}_o, \boldsymbol{\theta}) = \ell_\psi(\boldsymbol{x}_o|\boldsymbol{\theta})p(\boldsymbol{\theta})$ against the variational posterior $q_\phi$ (blue, obtained with the fKL), as well as the SIR-corrected density with $K = 2$ (orange) and $K = 32$ (green). Despite using an expressive normalizing flow as $q_\phi$, SIR improves the accuracy.

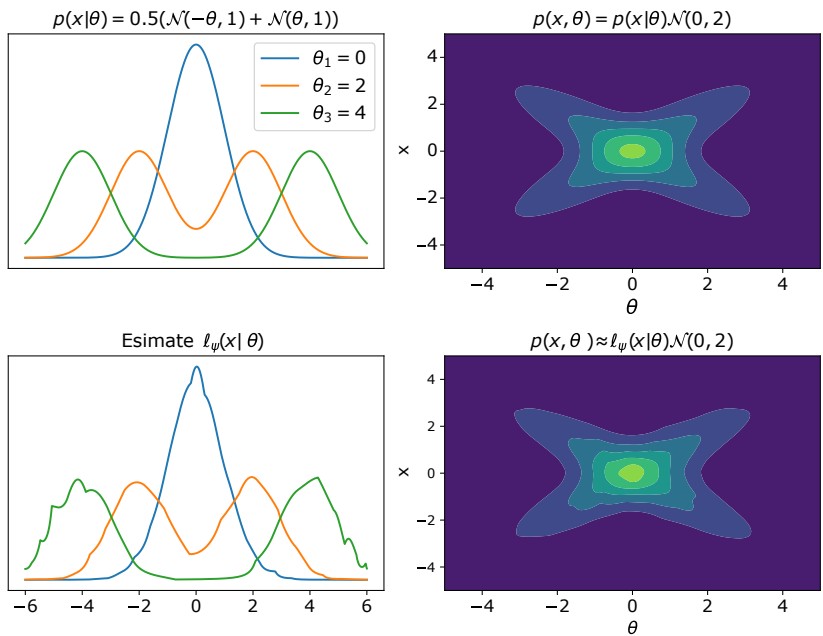

Figure 7: A neural spline flow (NSF) estimating a bimodal likelihood with $10^4$ simulations with prior $\mathcal{N}(0, 2)$. Top: Ground truth. Bottom: Likelihood-approximation with NSF. The learned likelihood closely matches the true likelihood.

### A.6 PROOFS FOR EXCLUDING INVALID DATA

Many simulators can produce unreasonable or undefined values when fed with parameters sampled from the prior (Lueckmann et al., 2017). These invalid simulations are not useful for accurately learning the likelihood, and we would like to ignore them. To do so, we developed a loss-reweighing strategy.

---

**Algorithm 3:** SNVI with calibration kernel

---

1   **Inputs:** prior $p(\boldsymbol{\theta})$, observation $\boldsymbol{x}_o$, divergence $D$, simulations per round $N$, number of rounds $R$, selection strategy $\mathcal{S}$ and calibration kernel $K$.

2   **Outputs:** Approximate likelihood $\ell_\psi$, variational posterior $q_\phi$ and calibration network $c_\zeta$.

3   **Initialize:** Proposal $\tilde{p}(\boldsymbol{\theta}) = p(\boldsymbol{\theta})$, simulation dataset $\mathcal{X} = \{\}$, calibration dataset $\mathcal{C} = \{\}$

4   **for** $r \in [1, ..., R]$ **do**

5       **for** $i \in [1, ..., N]$ **do**

6           $\boldsymbol{\theta}_i = \mathcal{S}(\tilde{p}, \ell_\phi, p)$ ;                          `// sample` $\boldsymbol{\theta}_i \sim \tilde{p}(\boldsymbol{\theta})$

7           simulate $\boldsymbol{x}_i \sim p(\boldsymbol{x}|\boldsymbol{\theta}_i)$ ;             `// run the simulator on` $\boldsymbol{\theta}_i$

8           add $(\boldsymbol{\theta}_i, K(\boldsymbol{x}_i, \boldsymbol{x}_o))$ to $\mathcal{C}$

9           **if** $K(\boldsymbol{x}, \boldsymbol{x}_o) > 0$ **then**

10              add $(\boldsymbol{\theta}_i, \boldsymbol{x}_i)$ to $\mathcal{X}$

11           **end**

12       **end**

13       (re-)train $\ell_\psi$;     $\psi^* = \arg\min_\psi -\frac{1}{N} \sum_{(\boldsymbol{x}_i, \boldsymbol{\theta}_i) \in \mathcal{X}} K(\boldsymbol{x}_i, \boldsymbol{x}_o) \log \ell_\psi(\boldsymbol{x}_i|\boldsymbol{\theta}_i)$ ;      `// or` `SNRE`

14       (re-)train $c_\zeta$;     $\xi^* = \arg\min_\xi \frac{1}{N} \sum_{(\boldsymbol{\theta}_i, K(\boldsymbol{x}_i, \boldsymbol{x}_o)) \in \mathcal{C}} \mathcal{L}(c_\zeta(\boldsymbol{\theta}_i), K(\boldsymbol{x}_i, \boldsymbol{x}_o))$ ;    `// MSE` `or cross-entropy for binary calibration kernel`

15       (re-)train $q_\phi$;     $\phi^* = \arg\min_\phi D(q_\phi(\boldsymbol{\theta}) || p(\boldsymbol{\theta}|\boldsymbol{x}_o))$ with

$$p(\boldsymbol{\theta}|\boldsymbol{x}_o) \propto p(\boldsymbol{x}_o|\boldsymbol{\theta})p(\boldsymbol{\theta}) \approx \ell_{\psi^*}(\boldsymbol{x}_o|\boldsymbol{\theta})c_{\xi^*}(\boldsymbol{\theta})p(\boldsymbol{\theta})$$

16       $\tilde{p}(\boldsymbol{\theta}) = q_\phi(\boldsymbol{\theta})$

17   **end**

---

We formulate the exclusion of invalid simulations by the means of a calibration kernel $K(\boldsymbol{x}, \boldsymbol{x}_o)$ (originally introduced for neural posterior estimation in Lueckmann et al. (2017)). This calibration kernel can be any function, and can thus be used beyond excluding invalid data. The case of excluding invalid simulations can be recovered by using a binary calibration kernel:

$$K(\boldsymbol{x}, \boldsymbol{x}_o) = \begin{cases} 0 & \text{if } \boldsymbol{x} \text{ invalid} \\ 1 & \text{if } \boldsymbol{x} \text{ valid} \end{cases} \tag{1}$$

Alg. 3 shows SNVI with calibration kernel. Notice that, in the case of a binary calibration kernel, the loss for the likelihood $\ell_\psi$;   $\psi^* = \arg\min_\psi -\frac{1}{N} \sum_{(\boldsymbol{x}_i, \boldsymbol{\theta}_i) \in \mathcal{X}} K(\boldsymbol{x}_i, \boldsymbol{x}_o) \log \ell_\psi(\boldsymbol{x}_i|\boldsymbol{\theta}_i)$ is zero for all invalid simulations (because $K(\boldsymbol{x}, \boldsymbol{x}_o) = 0$). Thus, since these simulations do not contribute to the loss, we exclude these simulations from the dataset that is used to train the likelihood(-ratio) model.

Below, we provide proofs of convergence for Alg. 3. Theorem 1 and Lemma 1 are relevant to SNLVI, Theorem 2 and Lemma 2 are relevant to SNRVI, and Lemma 3 is relevant to both methods. Theorem 1 and Theorem 2 provide a means to use a calibration kernel in the training of the likelihood(-ratio)-model such that one can still recover the posterior density. In SNVI, we sample from the (unnormalized) potential function with variational inference. However, one can also use Theorem 1 and Theorem 2 in combination with SNLE and SNRE and draw samples from the potential function with MCMC.

Both SNLVI and SNRVI with calibration kernels rely on the estimation of $\mathbb{E}_{\boldsymbol{x} \sim p(\boldsymbol{x}|\boldsymbol{\theta})}[K(\boldsymbol{x}, \boldsymbol{x}_o)]$ (note that this turns into $p(\text{valid}|\boldsymbol{\theta})$ for the binary calibration kernel). We estimate this term with a feed-forward regression neural network $c_\zeta(\boldsymbol{\theta})$ (see Lemma 3). The network is trained on pairs $(\boldsymbol{\theta}, K(\boldsymbol{x}, \boldsymbol{x}_o))$, where $\boldsymbol{\theta}$ and $\boldsymbol{x}$ are the same pairs as used for training the likelihood(-ratio)-model. For general calibration kernels $K(\boldsymbol{x}, \boldsymbol{x}_o)$, we use a mean-squared error loss, whereas in the case of invalid data, we parameterize $c_\zeta(\boldsymbol{\theta})$ as a logistic regression network and train it with a cross-entropy loss (since the calibration kernel $K(\boldsymbol{x}, \boldsymbol{x}_o)$ is a binary function: 0 for invalid data, 1 for valid data).

**Theorem 1.** *Let $K : \mathcal{X} \times \mathcal{X} \to \mathbb{R}^+$ be a kernel. Let $\ell_{\psi^*}(\boldsymbol{x}|\boldsymbol{\theta})$ be the maximizer of the objective*

$$\mathcal{L} = \mathbb{E}_{\boldsymbol{\theta}, \boldsymbol{x} \sim \tilde{p}(\boldsymbol{\theta}, \boldsymbol{x})}[K(\boldsymbol{x}, \boldsymbol{x}_o) \log(\ell_\psi(\boldsymbol{x}|\boldsymbol{\theta}))]$$

and let $c_{\zeta^*}(\boldsymbol{\theta})$ be the minimizer of

$$\mathcal{L} = \mathbb{E}_{\boldsymbol{\theta}, \boldsymbol{x} \sim \tilde{p}(\boldsymbol{\theta}, \boldsymbol{x})}[(c_\zeta(\boldsymbol{\theta}) - K(\boldsymbol{x}, \boldsymbol{x}_o))^2]$$

*Then the potential function*

$$\mathcal{P}(\boldsymbol{\theta}) = \ell_{\psi^*}(\boldsymbol{x}_o|\boldsymbol{\theta})p(\boldsymbol{\theta})c_{\zeta^*}(\boldsymbol{\theta})$$

*is proportional to the posterior density $p(\boldsymbol{\theta}|\boldsymbol{x}_o)$.*

*Proof.* Using Lemma 1 and Lemma 3, we get

$$\begin{aligned}
\mathcal{P}(\boldsymbol{\theta}) &= \ell_{\psi^*}(\boldsymbol{x}_o|\boldsymbol{\theta})p(\boldsymbol{\theta})c_{\zeta^*}(\boldsymbol{\theta}) \\
&= \frac{K(\boldsymbol{x}, \boldsymbol{x}_o)p(\boldsymbol{x}_o|\boldsymbol{\theta})}{\mathbb{E}_{\boldsymbol{x} \sim p(\boldsymbol{x}|\boldsymbol{\theta})}[K(\boldsymbol{x}, \boldsymbol{x}_o)]}p(\boldsymbol{\theta})\mathbb{E}_{\boldsymbol{x} \sim p(\boldsymbol{x}|\boldsymbol{\theta})}[K(\boldsymbol{x}, \boldsymbol{x}_o)] \\
&= K(\boldsymbol{x}, \boldsymbol{x}_o)p(\boldsymbol{x}_o|\boldsymbol{\theta})p(\boldsymbol{\theta}) \\
&\propto p(\boldsymbol{\theta}|\boldsymbol{x}_o)
\end{aligned}$$

$\square$

**Theorem 2.** *Let $K : \mathcal{X} \times \mathcal{X} \to \mathbb{R}^+$ be a kernel. Let $\ell_{\psi^*}(\boldsymbol{x}, \boldsymbol{\theta})$ be the minimizer of the objective*

$$\mathcal{L} = \mathbb{E}_{\boldsymbol{\theta}, \boldsymbol{x} \sim \tilde{p}(\boldsymbol{\theta}, \boldsymbol{x})}\left[K(\boldsymbol{x}, \boldsymbol{x}_o)\log(\ell_{\psi^*}(\boldsymbol{x}, \boldsymbol{\theta}))\right] + \mathbb{E}_{\boldsymbol{\theta}, \boldsymbol{x} \sim p(\boldsymbol{\theta})p(\boldsymbol{x})}\left[K(\boldsymbol{x}, \boldsymbol{x}_o)\log(1 - \ell_{\psi^*}(\boldsymbol{x}, \boldsymbol{\theta}))\right]$$

*and let $c_{\zeta^*}(\boldsymbol{\theta})$ be the minimizer of*

$$\mathcal{L} = \mathbb{E}_{\boldsymbol{\theta}, \boldsymbol{x} \sim \tilde{p}(\boldsymbol{\theta}, \boldsymbol{x})}[(c_\zeta(\boldsymbol{\theta}) - K(\boldsymbol{x}, \boldsymbol{x}_o))^2]$$

*Then the potential function*

$$\mathcal{P}(\boldsymbol{\theta}) = \ell_{\psi^*}(\boldsymbol{x}_o, \boldsymbol{\theta})p(\boldsymbol{\theta})c_{\zeta^*}(\boldsymbol{\theta})$$

*is proportional to the posterior density $p(\boldsymbol{\theta}|\boldsymbol{x}_o)$.*

*Proof.* Using Lemma 2 and Lemma 3, we get

$$\begin{aligned}
\mathcal{P}(\boldsymbol{\theta}) &= \ell_{\psi^*}(\boldsymbol{x}_o, \boldsymbol{\theta})p(\boldsymbol{\theta})c_{\zeta^*}(\boldsymbol{\theta}) \\
&= \frac{\mathbb{E}_{\boldsymbol{x} \sim p(\boldsymbol{x})}[K(\boldsymbol{x}, \boldsymbol{x}_o)]}{\mathbb{E}_{\boldsymbol{x} \sim p(\boldsymbol{x}|\boldsymbol{\theta})}[K(\boldsymbol{x}, \boldsymbol{x}_o)]}\frac{p(\boldsymbol{x}_o|\boldsymbol{\theta})}{p(\boldsymbol{x}_o)}p(\boldsymbol{\theta})\mathbb{E}_{\boldsymbol{x} \sim p(\boldsymbol{x}|\boldsymbol{\theta})}[K(\boldsymbol{x}, \boldsymbol{x}_o)] \\
&= \mathbb{E}_{\boldsymbol{x} \sim p(\boldsymbol{x})}[K(\boldsymbol{x}, \boldsymbol{x}_o)]\frac{p(\boldsymbol{x}_o|\boldsymbol{\theta})}{p(\boldsymbol{x}_o)}p(\boldsymbol{\theta}) \\
&\propto p(\boldsymbol{\theta}|\boldsymbol{x}_o)
\end{aligned}$$

$\square$

**Lemma 1.** *Let $K : \mathcal{X} \times \mathcal{X} \to \mathbb{R}^+$ be an arbitrary kernel. Then, the objective*

$$\mathcal{L} = \mathbb{E}_{\boldsymbol{\theta}, \boldsymbol{x} \sim \tilde{p}(\boldsymbol{\theta}, \boldsymbol{x})}[K(\boldsymbol{x}, \boldsymbol{x}_o)\log(\ell_\psi(\boldsymbol{x}|\boldsymbol{\theta}))]$$

*is maximized if and only if $\ell_\psi(\boldsymbol{x}|\boldsymbol{\theta}) = \frac{1}{Z(\boldsymbol{\theta})}K(\boldsymbol{x}, \boldsymbol{x}_o)p(\boldsymbol{x}|\boldsymbol{\theta})$ for all $\boldsymbol{\theta} \in support(\tilde{p}(\boldsymbol{\theta}))$, with normalizing constant $Z(\boldsymbol{\theta}) = \int K(\boldsymbol{x}, \boldsymbol{x}_o)p(\boldsymbol{x}|\boldsymbol{\theta})d\boldsymbol{x} = \mathbb{E}_{\boldsymbol{x} \sim p(\boldsymbol{x}|\boldsymbol{\theta})}[K(\boldsymbol{x}, \boldsymbol{x}_o)]$.*

*Proof.*

$$\begin{aligned}
\mathcal{L} &= \mathbb{E}_{\boldsymbol{\theta}, \boldsymbol{x} \sim \tilde{p}(\boldsymbol{\theta}, \boldsymbol{x})}[K(\boldsymbol{x}, \boldsymbol{x}_o)\log(\ell_\psi(\boldsymbol{x}|\boldsymbol{\theta}))] \\
&= \iint K(\boldsymbol{x}, \boldsymbol{x}_o)\tilde{p}(\boldsymbol{\theta}, \boldsymbol{x})\log(\ell_\psi(\boldsymbol{x}|\boldsymbol{\theta}))d\boldsymbol{x}d\boldsymbol{\theta} \\
&= \iint K(\boldsymbol{x}, \boldsymbol{x}_o)\tilde{p}(\boldsymbol{\theta})p(\boldsymbol{x}|\boldsymbol{\theta})\log(\ell_\psi(\boldsymbol{x}|\boldsymbol{\theta}))d\boldsymbol{x}d\boldsymbol{\theta} \\
&= \int \tilde{p}(\boldsymbol{\theta})\int K(\boldsymbol{x}, \boldsymbol{x}_o)p(\boldsymbol{x}|\boldsymbol{\theta})\log(\ell_\psi(\boldsymbol{x}|\boldsymbol{\theta}))d\boldsymbol{x}d\boldsymbol{\theta}
\end{aligned}$$

Since $\int K(\boldsymbol{x}, \boldsymbol{x}_o)p(\boldsymbol{x}|\boldsymbol{\theta})\log(\ell_\psi(\boldsymbol{x}|\boldsymbol{\theta}))d\boldsymbol{x} \propto -D_{\mathrm{KL}}(\frac{1}{Z(\boldsymbol{\theta})}K(\boldsymbol{x}, \boldsymbol{x}_o)p(\boldsymbol{x}|\boldsymbol{\theta}), \ell_\psi(\boldsymbol{x}|\boldsymbol{\theta}))$, this term is maximized if and only if $\ell_\psi(\boldsymbol{x}|\boldsymbol{\theta}) = \frac{1}{Z(\boldsymbol{\theta})}K(\boldsymbol{x}, \boldsymbol{x}_o)p(\boldsymbol{x}|\boldsymbol{\theta})$ for all $\boldsymbol{\theta} \in support(\tilde{p}(\boldsymbol{\theta}))$ with $Z(\boldsymbol{\theta}) = \int K(\boldsymbol{x}, \boldsymbol{x}_o)p(\boldsymbol{x}|\boldsymbol{\theta})d\boldsymbol{x} = \mathbb{E}_{\boldsymbol{x} \sim p(\boldsymbol{x}|\boldsymbol{\theta})}[K(\boldsymbol{x}, \boldsymbol{x}_o)]$. $\square$

**Lemma 2.** *Let $K : \mathcal{X} \times \mathcal{X} \to \mathbb{R}^+$ be an arbitrary kernel. Then, the objective*

$$\mathcal{L} = \mathbb{E}_{\boldsymbol{\theta},\boldsymbol{x} \sim \tilde{p}(\boldsymbol{\theta},\boldsymbol{x})} \left[ K(\boldsymbol{x},\boldsymbol{x}_o) \log(\ell_{\psi^*}(\boldsymbol{x},\boldsymbol{\theta})) \right] + \mathbb{E}_{\boldsymbol{\theta},\boldsymbol{x} \sim \tilde{p}(\boldsymbol{\theta})p(\boldsymbol{x})} \left[ K(\boldsymbol{x},\boldsymbol{x}_o) \log(1 - \ell_{\psi^*}(\boldsymbol{x},\boldsymbol{\theta})) \right]$$

*is minimized if and only if $\ell_\psi(\boldsymbol{x},\boldsymbol{\theta}) = \frac{\mathbb{E}_{\boldsymbol{x} \sim p(\boldsymbol{x})}[K(\boldsymbol{x},\boldsymbol{x}_o)]}{\mathbb{E}_{\boldsymbol{x} \sim p(\boldsymbol{x}|\boldsymbol{\theta})}[K(\boldsymbol{x},\boldsymbol{x}_o)]} \frac{p(\boldsymbol{x}|\boldsymbol{\theta})}{p(\boldsymbol{x})}$ for all $\boldsymbol{\theta} \in support(\tilde{p}(\boldsymbol{\theta}))$.*

*Proof.* We begin by rearranging the expectations:

$$\mathcal{L} = \mathbb{E}_{\boldsymbol{\theta},\boldsymbol{x} \sim \tilde{p}(\boldsymbol{\theta},\boldsymbol{x})} \left[ K(\boldsymbol{x},\boldsymbol{x}_o) \log(\ell_{\psi^*}(\boldsymbol{x},\boldsymbol{\theta})) \right] + \mathbb{E}_{\boldsymbol{\theta},\boldsymbol{x} \sim \tilde{p}(\boldsymbol{\theta})\tilde{p}(\boldsymbol{x})} \left[ K(\boldsymbol{x},\boldsymbol{x}_o) \log(1 - \ell_{\psi^*}(\boldsymbol{x},\boldsymbol{\theta})) \right]$$

$$= \iint \tilde{p}(\boldsymbol{\theta},\boldsymbol{x}) K(\boldsymbol{x},\boldsymbol{x}_o) \log(\ell_{\psi^*}(\boldsymbol{x},\boldsymbol{\theta})) d\boldsymbol{\theta} d\boldsymbol{x} +$$

$$\iint \tilde{p}(\boldsymbol{\theta})\tilde{p}(\boldsymbol{x}) K(\boldsymbol{x},\boldsymbol{x}_o) \log(1 - \ell_{\psi^*}(\boldsymbol{x},\boldsymbol{\theta})) d\boldsymbol{\theta} d\boldsymbol{x}$$

$$= \iint \frac{\tilde{p}(\boldsymbol{\theta},\boldsymbol{x}) K(\boldsymbol{x},\boldsymbol{x}_o)}{\iint \tilde{p}(\boldsymbol{\theta},\boldsymbol{x}) K(\boldsymbol{x},\boldsymbol{x}_o) d\boldsymbol{\theta} d\boldsymbol{x}} \log(\ell_{\psi^*}(\boldsymbol{x},\boldsymbol{\theta})) d\boldsymbol{\theta} d\boldsymbol{x} +$$

$$\iint \frac{\tilde{p}(\boldsymbol{\theta})\tilde{p}(\boldsymbol{x}) K(\boldsymbol{x},\boldsymbol{x}_o)}{\iint \tilde{p}(\boldsymbol{\theta})\tilde{p}(\boldsymbol{x}) K(\boldsymbol{x},\boldsymbol{x}_o) d\boldsymbol{\theta} d\boldsymbol{x}} \log(1 - \ell_{\psi^*}(\boldsymbol{x},\boldsymbol{\theta})) d\boldsymbol{\theta} d\boldsymbol{x}$$

$$= \mathbb{E}_{\boldsymbol{\theta},\boldsymbol{x} \sim \pi_{\text{joint}}(\boldsymbol{\theta},\boldsymbol{x})} \left[ \log(\ell_{\psi^*}(\boldsymbol{x},\boldsymbol{\theta})) \right] + \mathbb{E}_{\boldsymbol{\theta},\boldsymbol{x} \sim \pi_{\text{marginal}}(\boldsymbol{\theta},\boldsymbol{x})} \left[ \log(1 - \ell_{\psi^*}(\boldsymbol{x},\boldsymbol{\theta})) \right]$$

where we introduced

$$\pi_{\text{joint}}(\boldsymbol{\theta},\boldsymbol{x}) = \frac{\tilde{p}(\boldsymbol{\theta},\boldsymbol{x}) K(\boldsymbol{x},\boldsymbol{x}_o)}{\iint \tilde{p}(\boldsymbol{\theta},\boldsymbol{x}) K(\boldsymbol{x},\boldsymbol{x}_o) d\boldsymbol{\theta} d\boldsymbol{x}} \qquad \pi_{\text{marginal}}(\boldsymbol{\theta},\boldsymbol{x}) = \frac{\tilde{p}(\boldsymbol{\theta})\tilde{p}(\boldsymbol{x}) K(\boldsymbol{x},\boldsymbol{x}_o)}{\iint \tilde{p}(\boldsymbol{\theta})\tilde{p}(\boldsymbol{x}) K(\boldsymbol{x},\boldsymbol{x}_o)}$$

Since binary classification recovers density ratios (Cranmer et al., 2015; Mohamed & Lakshminarayanan, 2016; Gutmann et al., 2018), we get

$$\ell_{\psi^*}(\boldsymbol{x},\boldsymbol{\theta}) = \frac{\pi_{\text{joint}}(\boldsymbol{\theta},\boldsymbol{x})}{\pi_{\text{marginal}}(\boldsymbol{\theta},\boldsymbol{x})}$$

$$= \frac{\frac{1}{\iint \tilde{p}(\boldsymbol{\theta},\boldsymbol{x}) K(\boldsymbol{x},\boldsymbol{x}_o) d\boldsymbol{\theta} d\boldsymbol{x}} \tilde{p}(\boldsymbol{\theta},\boldsymbol{x}) K(\boldsymbol{x},\boldsymbol{x}_o)}{\frac{1}{\iint \tilde{p}(\boldsymbol{\theta})p(\boldsymbol{x}) K(\boldsymbol{x},\boldsymbol{x}_o) d\boldsymbol{\theta} d\boldsymbol{x}} \tilde{p}(\boldsymbol{\theta})\tilde{p}(\boldsymbol{x}) K(\boldsymbol{x},\boldsymbol{x}_o)}$$

$$= \frac{\iint \tilde{p}(\boldsymbol{\theta})\tilde{p}(\boldsymbol{x}) K(\boldsymbol{x},\boldsymbol{x}_o) d\boldsymbol{\theta} d\boldsymbol{x}}{\iint \tilde{p}(\boldsymbol{\theta})p(\boldsymbol{x}|\boldsymbol{\theta}) K(\boldsymbol{x},\boldsymbol{x}_o) d\boldsymbol{\theta} d\boldsymbol{x}} \frac{p(\boldsymbol{x}|\boldsymbol{\theta})}{\tilde{p}(\boldsymbol{x})}$$

$$= \frac{\int \tilde{p}(\boldsymbol{x}) K(\boldsymbol{x},\boldsymbol{x}_o) \int \tilde{p}(\boldsymbol{\theta}) d\boldsymbol{\theta} d\boldsymbol{x}}{\int p(\boldsymbol{x}|\boldsymbol{\theta}) K(\boldsymbol{x},\boldsymbol{x}_o) \int \tilde{p}(\boldsymbol{\theta}) d\boldsymbol{\theta} d\boldsymbol{x}} \frac{p(\boldsymbol{x}|\boldsymbol{\theta})}{\tilde{p}(\boldsymbol{x})}$$

$$= \frac{\mathbb{E}_{\boldsymbol{x} \sim \tilde{p}(\boldsymbol{x})}[K(\boldsymbol{x},\boldsymbol{x}_o)]}{\mathbb{E}_{\boldsymbol{x} \sim p(\boldsymbol{x}|\boldsymbol{\theta})}[K(\boldsymbol{x},\boldsymbol{x}_o)]} \frac{p(\boldsymbol{x}|\boldsymbol{\theta})}{\tilde{p}(\boldsymbol{x})}$$

$$\square$$

**Lemma 3.** *The objective*

$$\mathcal{L} = \mathbb{E}_{\boldsymbol{\theta},\boldsymbol{x} \sim \tilde{p}(\boldsymbol{\theta},\boldsymbol{x})}[(c_\zeta(\boldsymbol{\theta}) - K(\boldsymbol{x},\boldsymbol{x}_o))^2]$$

*is minimized if and only if $c_\zeta(\boldsymbol{\theta}) = \mathbb{E}_{\boldsymbol{x} \sim p(\boldsymbol{x}|\boldsymbol{\theta})}[K(\boldsymbol{x},\boldsymbol{x}_o))]$ for all $\boldsymbol{\theta} \in support(\tilde{p}(\boldsymbol{\theta}))$.*

*Proof.*

$$\mathcal{L} = \mathbb{E}_{\boldsymbol{\theta},\boldsymbol{x} \sim \tilde{p}(\boldsymbol{\theta},\boldsymbol{x})}[(c_\zeta(\boldsymbol{\theta}) - K(\boldsymbol{x},\boldsymbol{x}_o))^2]$$

$$= \iint \tilde{p}(\boldsymbol{\theta},\boldsymbol{x})(c_\zeta(\boldsymbol{\theta}) - K(\boldsymbol{x},\boldsymbol{x}_o))^2 d\boldsymbol{x} d\boldsymbol{\theta}$$

$$= \int \tilde{p}(\boldsymbol{\theta}) \int p(\boldsymbol{x}|\boldsymbol{\theta})(c_\zeta(\boldsymbol{\theta}) - K(\boldsymbol{x},\boldsymbol{x}_o))^2 d\boldsymbol{x} d\boldsymbol{\theta}$$

$$= \int \tilde{p}(\boldsymbol{\theta}) \mathbb{E}_{\boldsymbol{x} \sim p(\boldsymbol{x}|\boldsymbol{\theta})}[(c_\zeta(\boldsymbol{\theta}) - K(\boldsymbol{x},\boldsymbol{x}_o))^2] d\boldsymbol{\theta}$$

which is minimized if and only if $c_\zeta(\boldsymbol{\theta}) = \mathbb{E}_{\boldsymbol{x} \sim p(\boldsymbol{x}|\boldsymbol{\theta})}[K(\boldsymbol{x},\boldsymbol{x}_o))]$) for all $\boldsymbol{\theta} \in support(\tilde{p}(\boldsymbol{\theta}))$. $\square$

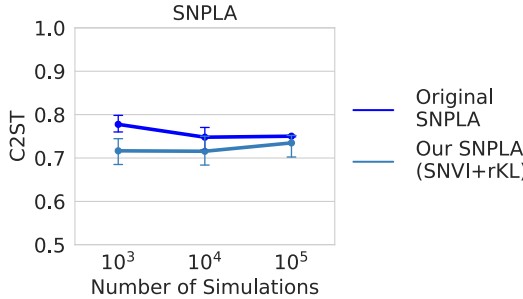

Figure 8: Comparison between SNPLA implementation of (Wiqvist et al., 2021) and SNVI with rKL.

## A.7 SNPLA

In Fig. 2 and Fig. 3, we compared SNVI to SNPLA (Wiqvist et al., 2021). To ensure comparability between SNPLA and SNVI, we implemented SNPLA ourselves and used the same likelihood- and posterior-model for both methods. The main difference between our implementation and the original implementation of SNPLA are:

1. We do not use the proposal $\hat{p}_r(\boldsymbol{\theta}) = \alpha p(\boldsymbol{\theta}) + (1-\alpha)q_\phi(\boldsymbol{\theta})$ for $\alpha \in [0,1]$, instead we use $\alpha = 0$, i.e. we use the current posterior estimate as proposal.

2. Secondly, we use a Rational Linear Spline Flow (RSF) based on pyro (Bingham et al., 2019), whereas Wiqvist et al. (2021) uses a Masked Autoregressive Flow based on nflows (Durkan et al., 2019b).

Fig. 8 compares the performance of our SNPLA implementation to the original implementation. Our implementation performs slightly better, likely due to the use of more expressive normalizing flows. We used our implementation for all experiments and nonetheless refer to the method with the name 'SNPLA'.

## A.8 EXPERIMENTS: BENCHMARK

All tasks were taken from an sbi benchmark (Lueckmann et al., 2021). For a description of the simulators, summary statistics, and prior distributions, we refer the reader to that paper.

We use the SNLE and SNRE as implemented in the sbi package (Tejero-Cantero et al., 2020). In all experiments, we learn the likelihood with a Masked Autoregressive Flow (MAF) with five autoregressive layers each with two hidden layers and 50 hidden units (Tejero-Cantero et al., 2020; Durkan et al., 2019b). For SNRE we use a two block residual network with 50 hidden units. Just as in Lueckmann et al. (2021), we implement SNRE with the loss described in Durkan et al. (2020).

The implementation of the posterior normalizing flows is based on pyro (Bingham et al., 2019), as pyro caches intermediate values during sampling and thus allow cheap density evaluation on obtained samples. We use MAFs for higher dimensional problems and Rational Linear Spline Flows (RSF) for low dimensional but complex problems (SLCP, Two moons). We always use a standard Gaussian base distribtuion and five autoregressive layers with a hidden size depending on input dimension ($[\dim \cdot 10, \dim \cdot 10]$ for spline autoregressive nets and $[\dim \cdot 5 + 5]$ for affine autoregressive nets, each with ReLU activations). As the posterior support must match that of the prior, we add a bijective mapping that maps the support to that of the prior. This allows to train the normalizing flows directly on the constrained domain.

We used a total sampling budget of $N = 256$ for any VI loss. To estimate the IW-ELBO we use $N = 32$ to estimate $\mathcal{L}_{IW}^{(K=8)}(\phi)$ (Rainforth et al., 2018). Additionally, we use the STL estimator (Roeder et al., 2017). An alternatively would be the doubly reparameterized gradient estimator, which is unbiased. We choose the STL estimator as it admits larger SNRs at the cost of introducing some bias (Tucker et al., 2018). Because for $\alpha \to 0$ we have that $\mathcal{L}_\alpha \to \mathcal{L}_{IW}^{(K=1)}$ we use this

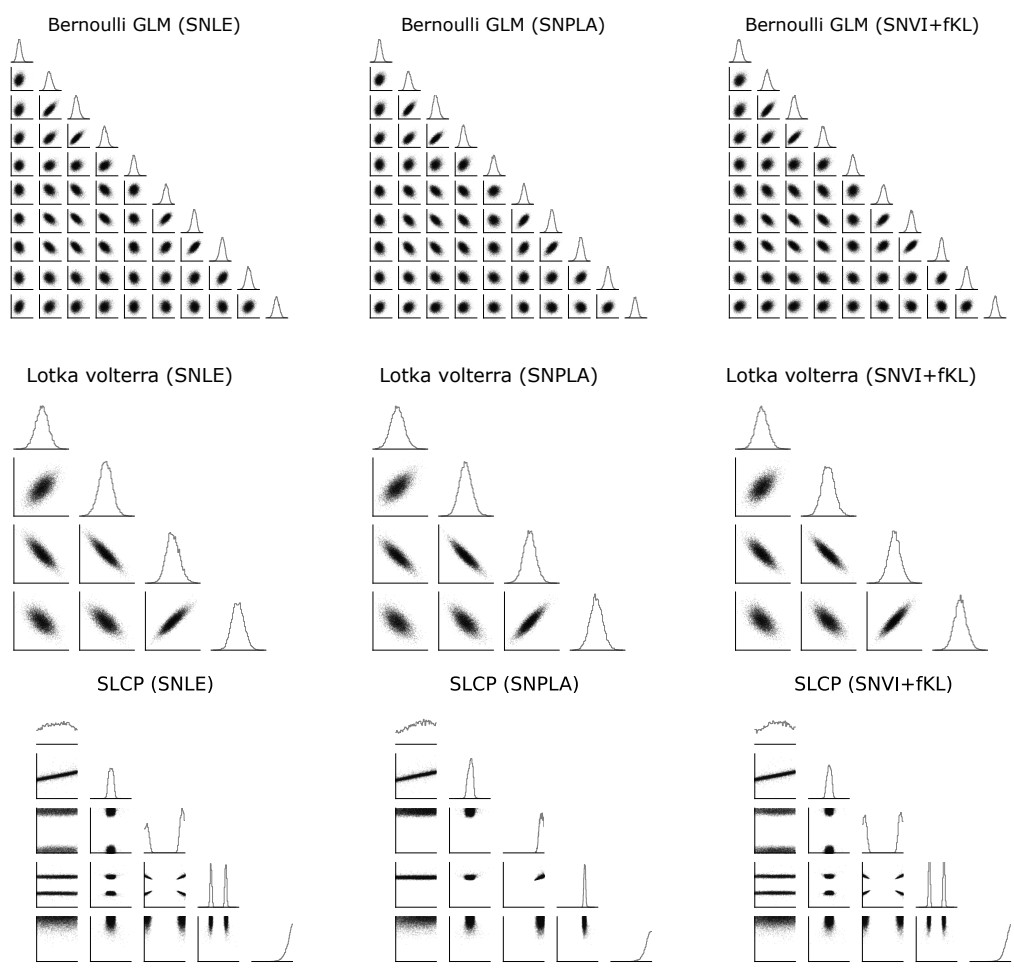

Figure 9: Samples from the posterior distributions for SNLE with MCMC, SNVI + fKL, SNVI + rKL. First row: results for SLCP. Second row: Lotka-Volterra. Third row: Bernoulli GLM.

estimator also to estimate $\mathcal{L}_{\alpha=0.1}(\phi)$. While the estimator can also be used for the ELBO, it requires additional computational cost i.e. we additionally need to calculate the inverse transformation, which is costly for autoregressive flows. Note that the fKL estimator also requires the inverse transform, thus we recommend to use a normalizing flow with fast forward and inverse passes in problems with many parameters, e.g. normalizing flows based on coupling layers (Dinh et al., 2017; Durkan et al., 2019a).

We trained for 10 rounds of simulations. In each round, we initialize the likelihood- and the posterior-model as their respective last estimates from the previous round. We train the posterior model for each round for at least 100 iterations and at most 1000 iterations. We evaluate convergence by tracking the decrease within the loss. For this automated benchmark, the convergence criteria are chosen conservative too avoid early stopping. More elaborate convergence criteria may improve runtime.

As metrics, we used classifier 2-sample tests (C2ST). C2ST trains a classifier to distinguish posterior samples produced by a specific method to ground truth posterior samples. Thus, a value of 0.5 means that the distributions are identical, whereas higher values indicate a mismatch between the distributions. As in Lueckmann et al. (2021), we computed the C2ST using 10,000 samples. Each figure shows the average metric value over 10 different observations, as well as the corresponding 95% confidence interval.

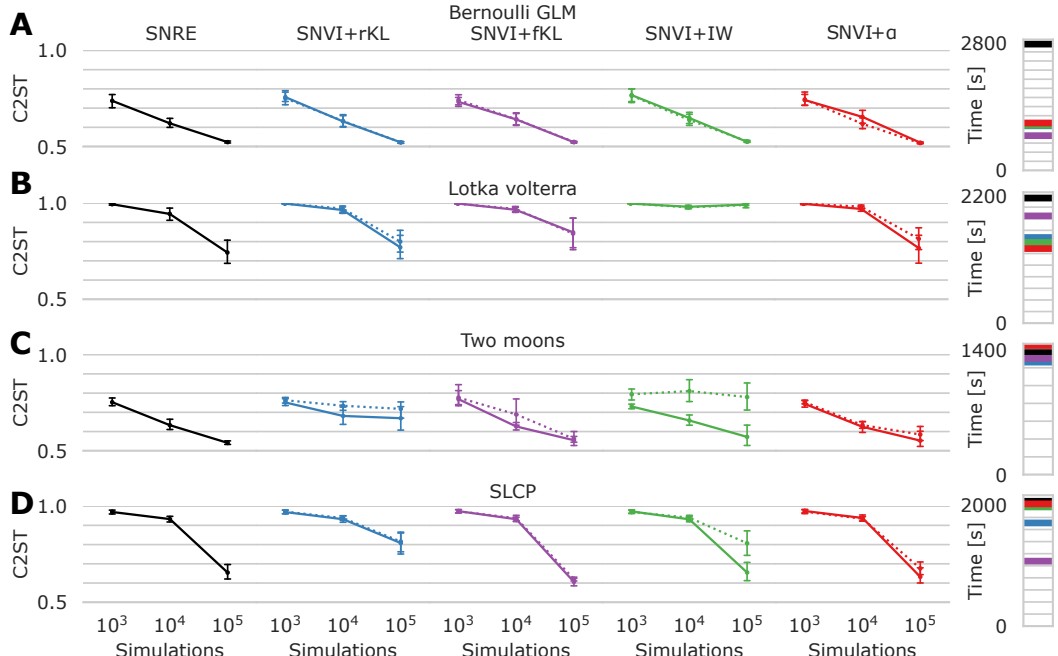

Figure 10: C2ST benchmark results for SNVI with ratio estimation (SNRVI) for four models, Bernoulli GLM (**A**), Lotka Volterra (**B**), Two moons (**C**) and SLCP (**D**). Each point represents the average metric value for ten different observations, as well as the confidence intervals. Bars on the right indicate the average runtime. Two reference methods: SNRE with MCMC sampling and the rKL, as well as three variants of SNVI, with forward KL (SNVI+fKL), importance-weighted ELBO (SNVI+IW) and $\alpha$-divergence (SNVI+$\alpha$). Dotted lines: performance when not using SIR.

Fig. 11 shows results for further variational objectives on the two moons (top) and on the SLCP task (bottom). The self-normalization used for the forward KL estimator improves the approximation quality (11, left, dark vs light purple). For the IW-ELBO (middle) as well as for the $\alpha$-divergences (right), the STL estimator improves performance (Rainforth et al., 2018). The gains from the STL are stronger for $\alpha$-divergences as for the IW-ELBO (especially when using SIR). The STL particularly improves the estimate for low values of $alpha$ (which are more support-covering).

## A.9    EXPERIMENTS: INFERENCE IN A NEUROSCIENCE MODEL OF THE PYLORIC NETWORK

We used the same simulator as in Gonçalves et al. (2020); Deistler et al. (2021) and the 15 summary statistics originally described in Prinz et al. (2004) and also used in Gonçalves et al. (2020); Deistler et al. (2021) (notably, Gonçalves et al. (2020); Deistler et al. (2021) used 3 additional features). Below, we describe the simulator briefly, for a full description we refer the reader to Prinz et al. (2004); Gonçalves et al. (2020); Deistler et al. (2021).

The model is composed of three single-compartment neurons, AB/PD, LP, and PY, where the electrically coupled AB and PD neurons are modeled as a single neuron. Each of the model neurons contains 8 currents. In addition, the model contains 7 synapses. As in Prinz et al. (2004), these synapses are simulated using a standard model of synaptic dynamics (Abbott & Marder, 1998).

For each set of membrane and synaptic conductances, we numerically simulate the circuit for 10 seconds with a step size of 0.025 ms. At each time step, each neuron receives Gaussian noise with mean zero and standard deviation 0.001 mV·ms$^{-0.5}$.

We applied SNVI to infer the posterior over 24 membrane parameters and 7 synaptic parameters, i.e. 31 parameters in total. The 7 synaptic parameters are the maximal conductances of all synapses in the circuit, each of which is varied uniformly in logarithmic domain and the membrane parameters

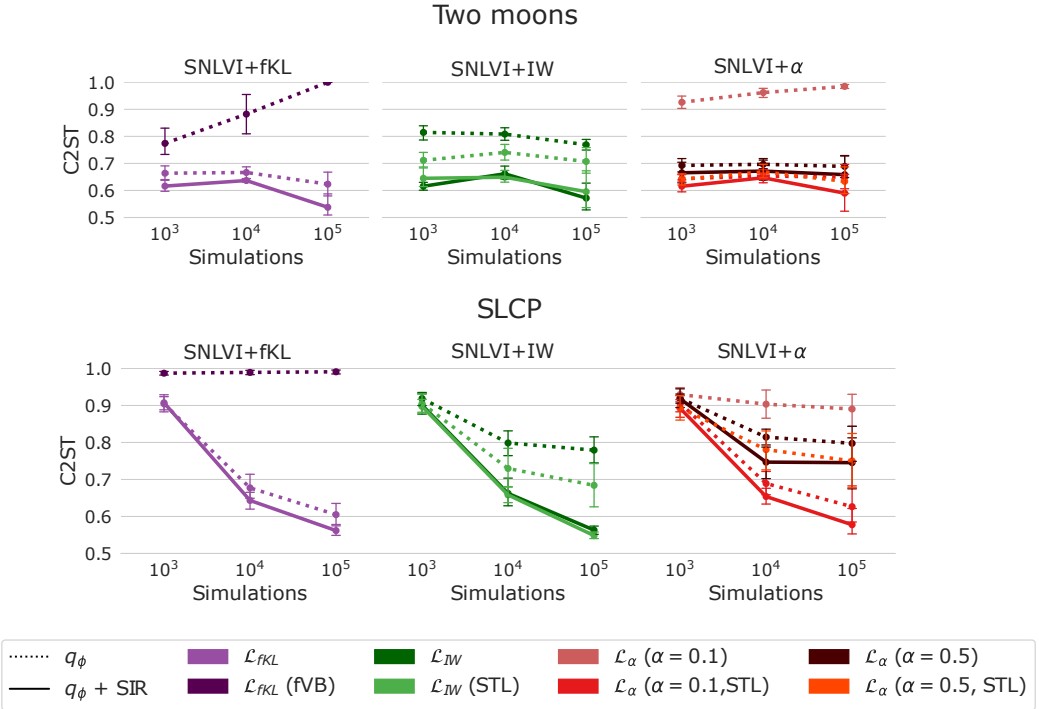

Figure 11: Evaluation of further variational objectives for the two moons (top) and the SLCP (bottom) task. Left: Variations of the forward KL (with and without self-normalized weights). Middle: Variations of the IW-ELBO (with and without STL). Right: Variations of the $\alpha$-divergence (with and without STL as well as for different values of $\alpha$.

are the maximal membrane conductances for each neuron. All membrane and synaptic conductances are varied over the same range as in Gonçalves et al. (2020); Deistler et al. (2021).

The 15 summary features proposed by Prinz et al. (2004) are salient features of the pyloric rhythm: Cycle period (s), three burst durations (s), two gap durations between bursts, two phase delays, three duty cycles, two phase gaps, and two phases of burst onsets. Note that several of these values are only defined if each neuron produces rhythmic bursting behavior. In particular we call any simulations invalid if at least one of the summary features is undefined.

The experimental data is taken from file 845_082_0044 in a publicly available dataset (Haddad & Marder, 2021).

For the likelihood-model, we use a Neural Spline Flow (NSF) with five autoregressive layers. Each layer has two hidden layers and 50 hidden neurons, as implemented in the sbi package (Tejero-Cantero et al., 2020; Durkan et al., 2019b). The posterior-model is a Masked autoregressive flow (MAF) with five autoregressive layers each with one hidden layer and 160 hidden units.

We train a total of 31 rounds. In the first round we use 50000 simulations from which only 492 are valid, and thus used to estimate the likelihood. For all other rounds we each simulated 10000 samples. To account for invalid summary features, we use the calibration kernel $K(\boldsymbol{x}, \boldsymbol{x}_o) = I(\boldsymbol{x}$ is valid$)$, hence can simply exclude any invalid simulations from training the likelihood-model. By Theorem 1 we have to correct the likelihood by multiplication of $\mathbb{E}_{\boldsymbol{x} \sim p(\boldsymbol{x}|\boldsymbol{\theta})}[I(\boldsymbol{x}$ is valid$)] = P(\boldsymbol{x}$ is valid$|\boldsymbol{\theta})$. To estimate this probability we use a deep logistic regression net with 3 hidden layers each with 50 neurons and ReLU activations. We train this classifier simultaneously with the likelihood-model, that is in each round we add new data $\{(\boldsymbol{\theta}_i, I(\boldsymbol{\theta}_i$ is valid$))\}_{i=1}^N$ and retrain the classifier using the weighted binary-cross-entropy loss. We weight the loss by the estimated class probabilities to account for class imbalance especially in early rounds. We fix the number of epochs to 200 per round. We use the fKL loss with $N = 1024$ samples, as well as SIR.

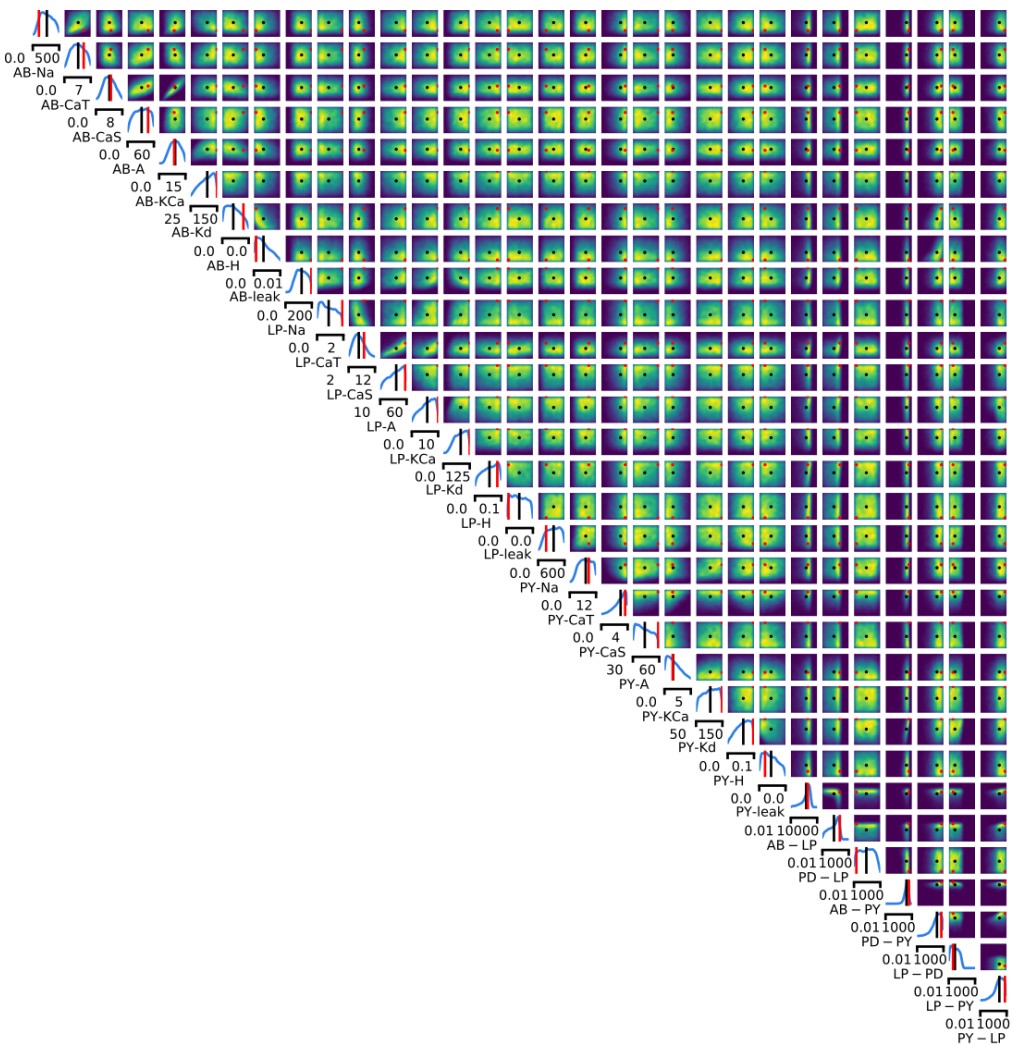

Figure 12: Posterior distribution for the neuroscience model of the pyloric network. In Fig. 4B we show a subset. The black point is a mean estimate using $10^7$ samples. The red point is a maximum a-posterior estimate, obtained by gradient ascent.

In total, the procedure took 27 hours, with the runs of the simulator being parallelized across several nodes. Because of this, the runtime also depends greatly on availability of computing resources on the cluster.

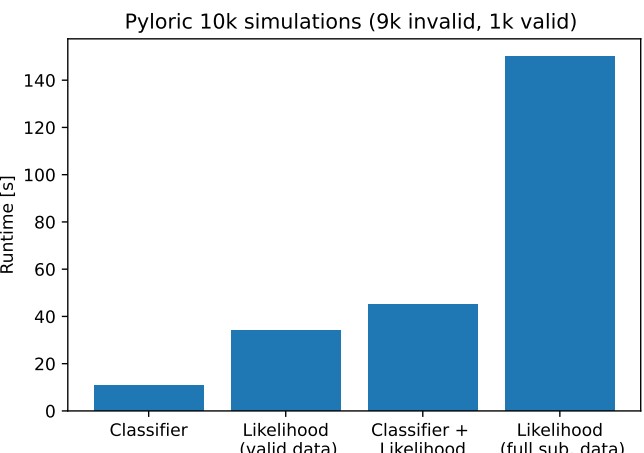

Figure 13: Runtime of the classifier $c_\zeta(\boldsymbol{\theta})$ in the model of the pyloric network (90% of simulations are invalid). Training the classifier is approximately three times cheaper than training the likelihood-model (compare left bar to second left) and thus increases the computational cost only modestly. The likelihood-model is trained only on valid simulations. The combined runtime of classifier and likelihood-model (third bar) is still far less then the time it would take to train the likelihood-model on all simulations (right bar. To estimate the runtime of the likelihood-model on all simulations, we substituted invalid simulation outputs (i.e. NaN) with an unreasonably low value and trained on all simulations).

