# OpenReview forum: "Variational methods for simulation-based inference"
_ICLR.cc/2022/Conference — ICLR 2022 Spotlight_

### Official Review · Reviewer_JNHt · 2021-11-02

**Correctness:** 4
**Technical Novelty And Significance:** 2
**Empirical Novelty And Significance:** 3
**Recommendation:** 8
**Confidence:** 4

**Main Review:**

Strengths:
- This work solves a quite practical problem of SNLE and SNRE.
- Experimental results show convincing results. Accuracy is maintained while reducing the computational cost of inference.
- Experiments against SNPLA demonstrate that using any of three proposed variational objectives leads to better results. Modes in the posterior approximations are now correctly captured.
- Posterior predictive checks are used to evaluate the quality of the approximate posteriors.
- The supplementary materials are thorough and include helpful experimental details as well as further discussions.

Weaknesses:
- Figure 8 is meant to show results for SNRE and SNRVI, but the labels state SNLE and SNVI. Are the labels wrong?
- Novelty is limited in comparison to SNPLA, but this is not a big issue as far as I am concerned since SNVI improves upon SNPLA. It now works empirically much better.

Questions:
- The self-normalized importance sampling used to estimate fKL requires the evaluation of $w(\theta)=p(x_o, \theta)/\pi(\theta)$. How is $p(x_o, \theta)$ computed, since we are in the likelihood-free setup?

**Summary Of The Paper:**

This paper presents SNVI, a simulation-based inference based likelihood (ratio) estimation and variational inference. The main contribution is to sidestep the slow MCMC sampling step of SNLE (SNRE) by learning the posterior distribution with variational inference. This work is similar to SNPLA but replaces the rKL variational objective with three alternative objectives that induce a mass-covering behaviour and are, therefore, better suited when using the approximate posterior distributions as proposal distributions in the sequential regime. Experiments show convincing results.


**Summary Of The Review:**

This paper presents a simple and effective idea. It makes both SNLE and SNRE more computationally efficient while maintaining accuracy. Experiments demonstrate this clearly. Provided that my question above can be answered, I recommend this paper for acceptance.

---

> ### Author Response · Authors · 2021-11-13
> **Response to reviewer JNHt**
>
> We thank the reviewer for their insightful comments and feedback! We are glad that the reviewer acknowledges the relevance, “convincing” results, and evaluations of our work. We do hope that the revised version of the manuscript (new supplementary figures and several extensions to the manuscript) will allow them to provide a stronger recommendation for acceptance. Alternatively, we would appreciate it if they could give us more direct guidance for which modifications or explanations would allow them to provide a stronger recommendation.
>
> > Figure 8 is meant to show results for SNRE and SNRVI, but the labels state SNLE and SNVI. Are the labels wrong?
>
> Yes, apologies! Thanks for detecting this issue, it will be fixed.
>
> > Novelty is limited in comparison to SNPLA, but this is not a big issue as far as I am concerned since SNVI improves upon SNPLA. It now works empirically much better.
>
> We acknowledge that SNLVI is similar to SNPLA (as described in the paper)-- but by identifying its shortcomings and improving upon them, it leads to a method that indeed performs vastly better empirically. We appreciate that you agree that this marked improvement in performance is important. We emphasize that our results show vastly improved runtime while maintaining accuracy compared to SNLE. Thus, users no longer face a trade-off as to whether they prefer low runtime (SNPLA) or good performance (SNLE): With SNVI, they can have the best from both worlds. We therefore expect that SNVI can become the go-to approach in simulation-based inference.
>
> > The self-normalized importance sampling used to estimate fKL requires the evaluation of w(θ)=p(xo,θ)/π(θ). How is p(xo,θ) computed, since we are in the likelihood-free setup?
>
> We use the surrogate likelihood $p(x_o,\theta) \approx l(x_o|\theta)p(\theta)$.

---

> > ### Comment · Reviewer_JNHt · 2021-11-19
> > **Thanks!**
> >
> > Thanks for your answer and clarifications. I confirm my positive evaluation. In light of your answers to the other reviews and the changes you committed doing to further improve the manuscript, I have increased my evaluation.

---

### Official Review · Reviewer_9Rvr · 2021-11-03

**Correctness:** 4
**Technical Novelty And Significance:** 2
**Empirical Novelty And Significance:** 2
**Recommendation:** 6
**Confidence:** 4

**Main Review:**

Strengths:

+ The idea is natural and the explanation is clear.

+ It improves efficiency comparing to MCMC based SNLE and SNRE while achieves similar accuracy.

+ The paper does a relatively complete ablation study. It compares the two types of KL divergence, MCMC and VI, and different variational objective in this setting.


Weaknesses

- The novelty of the paper is not very high. The major difference between the proposed method and  standard variational inference algorithm (and its variant) is that the likelihood here is not explicit.  The paper proposes to replace the unknown likelihood with an estimated likelihood, which is natural but the novelty is not very high.

- The paper propose several methods to improve the inference quality, e.g., SIR, IWAE, $\alpha$-divergence, but the reviewer concerns more about the accuracy in likelihood estimation. Can the  likelihood estimation capture the multimodality and skewness? Is there a simulation to demonstrate this accuracy?

- The calibration kernel in Sec. 3.4 is not explained clearly. How to learn $c(\theta)$? how to draw sample from (unnormalized) posterior distribution? Does the simulations in Sec. 4.1 and 4.2 use the calibration kernel? Why the Algorithm 1 doesn’t have calibration kernel?

- One question is that in the first step, instead of learning $p(x|\theta)$ from $(x, \theta)$ pairs, can we just learn $p(\theta | x)$ from $(x, \theta)$ pairs and use it to estimate $\theta$? How does the proposed method compare to this simple approach?

- The notations of the paper are sometimes confusing; here are some example:
In the Figure 1, the KL divergence is backward but in the method description, the KL is forward.
On page 4, should  $q_\psi$ be $q_\phi$?
On page 4, what are the subscripts of \theta in the equation?



**Summary Of The Paper:**

This paper proposes Sequential Neural Variational Inference (SNVI) for the posterior inference when the likelihood function is unknown. The proposed methods consist of a likelihood estimator, a posterior approximation, and sampling importance resampling.


**Summary Of The Review:**

In sum, this paper proposes a practical way for the posterior inference with unknown likelihood. The method is simple and natural, the writing is mostly clear. However, some sections and notations are not clear, and it is better to show the limitation of each component of the proposed method.

---

> ### Author Response · Authors · 2021-11-13
> **Response to reviewer 9Rvr**
>
> We thank the reviewer for their insightful comments, suggestions, and feedback. We are happy that the reviewer finds our idea natural and clearly explained, and acknowledges the resulting improvement in computational efficiency, as well as our efforts to do extensive ablation studies.
>
> > The major difference between the proposed method and standard variational inference algorithm (and its variant) is that the likelihood here is not explicit. The paper proposes to replace the unknown likelihood with an estimated likelihood [...].
>
> We emphasize that simulation-based inference methods are used extensively in scientific applications (Cranmer et al 2020 PNAS), and that our approach is substantially more simulation-efficient than currently used approaches (see R NxCC and JNHt), so that we expect it to have potential to become a go-to method for SBI. We acknowledge that SNVI is built from a careful and targeted combination of several components (each of which, individually, existed beforehand and is clearly described). In combination, these methods lead to massively improved efficiency compared to previous methods. But we emphasize that we do not simply “replace” the unknown likelihood with an "estimated" one, but rather that we estimate the likelihood ‘on the fly’ and we adaptively decide where to simulate and estimate. In previous likelihood(ratio) methods, each decision on where to sample required MCMC, which is _not_ the case in our approach (as the posterior estimate, which is used as the basis for deciding where next to simulate, is updated with VI).
>
> > The paper propose several methods to improve the inference quality, [...], but the reviewer concerns more about the accuracy in likelihood estimation. Can the likelihood estimation capture the multimodality and skewness? Is there a simulation to demonstrate this accuracy?
>
> The likelihood is estimated by a normalizing flow, which can capture the multimodality and skewness of a large variety of distributions - this has already been demonstrated in previous work (see e.g. Papamakarios et al. 2019 AISTATS, and Lueckmann et al AISTATS 2021 for a benchmark), but we will add a figure to the supplementary that shows that we can learn multimodal likelihoods. Finally, we point out that some of the demonstrations are based on tasks which have been established in previous work precisely because they can only be solved adequately with likelihood-estimates which capture skewness and multimodality.
>
> > The calibration kernel in Sec. 3.4 is not explained clearly [...]
>
> We acknowledge that our motivation for how the calibration kernel is used and the description of how it is technically implemented was not optimal, and we will include a more detailed explanation in the final paper. In our response to Reviewer NhK3, we outline the motivation behind the calibration kernel in more detail. As outlined in the appendix, we learn $c(\theta)$ with a feedforward neural network. The unnormalized posterior is sampled with variational inference (using either of the three described divergences + SIR). The simulations within the benchmark (section 4.1 and 4.2) do not use the calibration kernel because these toy examples do not produce invalid simulations (NaN, inf). We did not include it within Algorithm 1 as it only becomes necessary if simulations are invalid. We will provide a description of the entire algorithm (with calibration kernel) in the appendix and provide a clearer description of the issues mentioned by the reviewer in the paper.
>
> > [...] instead of learning p(x|θ) from (x,θ) pairs, can we just learn p(θ|x) from (x,θ) pairs and use it to estimate θ? How does the proposed method compare to this simple approach?
>
> This approach is known as (Sequential) Neural Posterior Estimation (SNPE) in the literature. A key disadvantage of SNPE is that it requires adjustments to the loss function when the parameters are not sampled from the prior but using some active learning scheme (in our example, by drawing parameters from the previous estimate of the posterior). In addition, SNPE does not allow ‘recycling’ of likelihoods across inference tasks. In the model in Sec 4.3, we show that SNVI, but not SNPE, can lead to good empirical results.
>
> > The notations of the paper are sometimes confusing [...]?
>
> We apologize-- In Fig1 (below the posterior approximation) we denote a general divergence measure with D, as we introduce in Sec 2.2. We will clarify this in the figure caption. The distinction between “forward” and “reverse” within the arguments (i.e. D(q||p) vs D(p||q)) depends on the definition of the divergence measure. We denote the KL divergence by $D_{KL}$ to make this distinction. Thanks for detecting the typo on page 4, we will fix it! We use subscripts on thetas to indicate independent but identically distributed samples. We will clarify this.

---

> > ### Comment · Reviewer_9Rvr · 2021-11-19
> > **Thanks for the authors' feedback**
> >
> > The authors address my major concerns and promises further revision. I would lean my recommendations towards acceptance.

---

> > ### Public Comment · ~Kirin_Ciao1 · 2025-05-20
> > **The notations in the algorithm 1 and 3**
> >
> > Should $\ell_\phi$ be $\ell_\psi$ in line 6 of algorithm 1 and 3?

---

### Official Review · Reviewer_NxCC · 2021-11-04

**Correctness:** 3
**Technical Novelty And Significance:** 3
**Empirical Novelty And Significance:** 3
**Recommendation:** 8
**Confidence:** 3

**Main Review:**

## Pros:
- The paper is well written.
- Improving the efficiency of sequential SBI methods is important.
- The paper nicely combines advanced methods and the results are good.
## Cons:
- It is yet not clear on which occasion it is a better idea to use a sequential method that will only be used once to do inference than an amortized method. I am sure there are good motivations for this but this is probably something that should motivate.
- By using SIR it looks like you trust more the likelihood surrogate than the approx. posterior. I do not see any strong motivations for this in the general case where both distributions may be complex.
- How does the method compare to simply learning an approximate posterior with a flow by minimizing some KL. It goes back to my first point, in which cases building the posterior online becomes more efficient.
- There are now many SBI methods and I think this would make the reader's life way easier if you (at least briefly) introduce the methods you compare your algorithm to and why you do not compare to the many other methods.
- Maybe you could give some additional context to the IW-ELBO (being explicit that it reduces the GAP between ELBO and likelihood).
- It is not clear to me what should a user do from the 3 possible losses to optimize phi.
- As you say that your method could be useful when we have many observations it would be interesting to show how does something that other methods cannot do.

**Summary Of The Paper:**

This paper presents Sequential Neural Variational Inference (SNVI), a new simulation-based inference (SBI) technique. SNVI sequentially updates a surrogate likelihood model and an approximate posterior model in high posterior density. Finally, the approximate posterior is sampled with importance resampling. Experiments shows this technique is able to learn multimodal posterior distributions and is faster than methods relying on MCMC for generating posterior samples while being almost as accurate.

**Summary Of The Review:**

I recommend weak acceptance as I think the method described in this paper may be interesting for the SBI community. I also find the paper well written and experiments convincing.

---

> ### Author Response · Authors · 2021-11-13
> **Response to reviewer NxCC**
>
> We thank the reviewer for the constructive feedback and comments! We are glad that they found it to be “well-written” and that they acknowledge the importance of improving computational efficiency of SBI methods, and that our paper ‘nicely combines advanced methods’ and that the ‘results are good’. We provide further explanations regarding the usefulness of sequential methods, the usage of SIR, and comparisons to other methods.
>
> > It is yet not clear on which occasion it is a better idea to use a sequential method that will only be used once to do inference than an amortized method.
>
> Providing a general answer on when to use sequential methods and when not was not the goal of the paper, but is discussed elsewhere in the literature: Non-sequential methods can be useful when one wants to amortize over many inference tasks (Cranmer et al PNAS 2020), but for doing inference on specific tasks, an empirical comparison by Lueckmann et al. 2021 (AISTATS) showed that sequential methods are more simulation-efficient than amortized methods. This is also one of the main reasons that SNVI requires 22 times fewer simulations than NPE on the pyloric network model (Sec 4.3). We will provide a more clear statement in the final version of the paper.
>
> > By using SIR it looks like you trust more the likelihood surrogate than the approx. posterior.
>
> The optimal variational posterior will fit the approximate posterior as defined by the surrogate likelihood $p(\theta|x) \propto l(x|\theta)p(\theta)$-- thus, the surrogate posterior is ‘twice’ approximated, and is generally therefore expected to be less accurate than the surrogate likelihood. The usefulness of the SIR step is demonstrated in Fig3 (dotted vs solid lines). We will add an additional supplementary figure that demonstrates why SIR works.
>
> > How does the method compare to simply learning an approximate posterior with a flow by minimizing some KL.
>
> This approach exists, and is known as (Sequential) Neural Posterior Estimation (SNPE). A key disadvantage of SNPE is that it requires adjustments to the loss function when the parameters are not sampled from the prior but using some active learning scheme. In addition, SNPE does not allow ‘recycling’ of likelihoods across inference tasks. In the pyloric network model, we show that SNVI, but not SNPE, can lead to good empirical results (Sec 4.3).
>
> > There are now many SBI methods and I think this would make the reader's life way easier if you [...] introduce the methods you compare your algorithm to and why you do not compare to the many other methods.
>
> We briefly describe all common neural-network based approaches in the Introduction and Background sections. We use the same tasks as in Lueckmann et al. 2021 (AISTATS) and compared to SOTA algorithms on benchmark tasks-- this will allow the reader to compare the performance of several other methods and even facilitate comparison with future SBI methods. We will explain this more clearly in the paper.
>
> > Maybe you could give some additional context to the IW-ELBO (being explicit that it reduces the GAP between ELBO and likelihood).
>
> This is indeed the main reason to use the IWELBO within VAE, as there one uses the lower bound as surrogate for the marginal likelihood to train the generative model. We do not emphasize this property, as, in a VI setting, one does not use the lower bound as a surrogate for the marginal likelihood. Nonetheless, we may provide this context too as it is the most common reasoning behind using the IW-ELBO in the VAE setting.
>
> > It is not clear to me what should a user do from the 3 possible losses to optimize phi.
>
> We should have been more clear-- Our results suggest that each of the three losses performs very similarly in terms of accuracy and efficiency (and much better than mode-seeking approaches). As we can not exclude the possibility that their performance might differ for other tasks, we chose to describe all three variants-- however, (see also responses above) we realize that this comes across as confusing. We will add a paragraph in the paper that discusses different aspects of each method and that will allow users to make an informed choice for the specific problem at hand (taking into account computation time and architecture of the normalizing flow).
>
> > As you say that your method could be useful when we have many observations it would be interesting to show how does something that other methods cannot do.
>
> As our method learns an explicit surrogate of the likelihood we can obtain the approximate unnormalized posterior by multiplying the likelihood of all datapoints: $p(x_1, \dots, x_n,\theta) = \prod_{i=1}^N p(x_i|\theta) p(\theta) \approx \prod_{i=1}^N l_\psi(x_i|\theta) p(\theta)$. This property is, for example, extremely useful and exploited in applications in particle physics (Cranmer, Pavez, Louppe, 2016). We will provide a citation in support of the importance of this point in the manuscript.

---

> > ### Comment · Reviewer_NxCC · 2021-11-14
> > **Raised score to accept**
> >
> > I thanks the authors for addressing my comments. I mostly agree with the authors’ response and have thus increase my score to 8.

---

### Official Review · Reviewer_NhK3 · 2021-11-07

**Correctness:** 4
**Technical Novelty And Significance:** 2
**Empirical Novelty And Significance:** 3
**Recommendation:** 6
**Confidence:** 3

**Main Review:**

The paper is clear from a method-recipe perspective. It has communicated well what the key aspects and elements of the approach are and how they are used together to form the overall approach.

The main weakness of the proposed approach is that it takes the form of combining multiple existing techniques into one coherent approach. There is indeed value to the community in formulating an approach that blends the strengths in multiple existing techniques together to overcome their individual weaknesses. However, both the techniques themselves and the way they are orchestrated together are not particularly novel or insightful. Specifically, each of the three parts to the SNVI approach, as well as the three variants of the variational objective used for the posterior model part of the approach, the SIR step, and the fix using calibration kernel, have already been well established by previous works, which the authors do clearly cite. Again, there is indeed value in orchestrating these techniques together, however in its current form the way this is done is neither particularly elegant or insightful.

As a minor and related weakness, it is not clear how one would choose between the three variants of the variational objective here. In terms of writing, each of these are briefly discussed with a summary and citation to the original work without much discussion on motivating or explaining them in relation to the wider proposed approach. I also find the discussion on SIR lacking and difficult to understand exactly how this enriches the variational family, although there is a citation to the original source which may have explained it.

Finally, the proposed method to get around missing simulation using the calibration kernel seems arbitrary and over-convoluted. This introduces not just a new set of hyperparameters that is not part of the main approach to worry about, but also an entire third model on top of the likelihood(-ratio) and posterior models.

To change my sentiment, I would find it most helpful if the authors could provide some explanations or arguments that would incline me to believe that:
- The choices made in this paper must be done in the particular way described for it to have the desired properties, and is not arbitrary.
- There are alternative choices for each component of this technique that seem sensible but can be shown to be worse than this combination of techniques.
- The method without SIR is not sufficient ensure the quality of the posterior samples despite expressiveness of NFs (as well as how the SIR step works, ideally in pseudo-code)
- The calibration kernel fix and the extra bias-factor model is not arbitrary and its learning or selection does not bear too much additional computational hurdle to the overall technique

-----
After author response:

Thank you for your response and clarifications, and especially on acting on the feedback.

The authors has addressed my concerns in detail and communicated how they would revise the paper accordingly for the camera-ready version. I would lean towards acceptance.

**Summary Of The Paper:**

This paper presents a form of variational inference (VI) approach for likelihood-free inference (LFI) named Sequential Neural Variational Inference (SNVI). Its main advantage over other VI approaches for LFI is computational efficiency rather than accuracy.

This approach has consists of three parts, a likelihood or likelihood-ratio model, a posterior model, and a sampling importance resampling (SIR) step to refine the posterior model each round. Because the variational family for the posterior model used is based on normalising flows (NFs), whose expressiveness is already very high, the authors instead focus on the variational objective of the posterior model. Here they consider three variants from previous works - (1) forward KL divergence with self-normalized importance sampling, (2) importance weighted ELBO, and (3) Rényi $\alpha$-divergences. Method (1) prevents objective weights to be close to zero by normalising them, at the cost of introducing a bias that vanishes at rate $\mathcal{O}(1/N)$. Method (2) aims to provide good proposals for SIR through lower bounding the ELBO, and uses the "Sticking the Landing" (STL) estimator to circumvent the low signal-to-noise (SNR) ratio. Method (3) allow the mass-covering or model-seeking behaviour to be tuned, and require the same STL estimator to prevent low SNR.

Finally, the authors propose an SIR step from previous work which claims to enrich the variational family with minimal computational cost.

In the case of invalid data from simulations, a kernel is used to reweigh the loss to focus on specific regions in the data-space. To fix the bias, the bias factor is estimated from another model to recover the posterior up to proportionality.


**Summary Of The Review:**

Overall, while the technique proposed seems useful in improving computational efficiencies of existing VI approaches to LFI, the technique itself is a relatively uninspiring combination of existing techniques, of which a few aspects thereof seem either lacking in explanation or over-convoluted in construction. I would tend towards a reject, but welcome the authors to challenge this sentiment by addressing the points in my main review.

---

> ### Author Response · Authors · 2021-11-13
> **Response to reviewer NhK3**
>
> We thank the reviewer for their constructive comments and the clear summary that indicates that they took the time to read the manuscript in depth and engaged with its content. We are glad that the reviewer finds the paper “well communicated” and that the approach “improves computational efficiency”. We hope that the revised version of the manuscript will convince the reviewer that there is a clear rationale behind our modelling choices and that each component makes a specific and necessary contribution, leading to a substantial improvement over the SOTA.
>
> We agree that SNVI carefully combines multiple existing components in a targeted fashion. Indeed, we see this as a compliment rather than a weakness: Our goal was performance and practical utility, and we believe SNVI achieves that goal (as shown by our empirical results and e.g. R JNHt). We identified three variational objectives which empirically lead to good performance-- we emphasize that all of our performance benefits over previous methods hold for all three approaches-- but we realize that describing them ‘side by side’ might have been confusing. We will add a paragraph that discusses different aspects of each method and will allow users to make an informed choice for the specific problem at hand (taking into account computation time of the simulator and architecture of the norm flow).
>
> > The choices made in this paper must be done in the particular way [...] and is not arbitrary.
>
> 1) We use Likelihood(-ratio) estimation, as the correction steps required when using Posterior estimation (SNPE) can be problematic e.g. by leading to leakage (see Durkan et al. 2020 ICML, also Sec 4.3). Fig4 clearly shows an example where SNVI is preferable to SNPE.
> 2) We use mass-covering divergences in order to avoid mode-collapse on multi-modal posteriors (often encountered in sbi problems). Fig2+3 clearly demonstrate the benefits of using mass-covering divergences over mode-seeking divergences.
> 3) The SIR step consistently and strongly improves the accuracy of the posterior (compare dotted line to solid line in Fig3, see also Agrawal et al 2020). We will add further results for **why** SIR works to the paper (see below).
>
> > There are alternative choices for each component of this technique that seem sensible but can be shown to be worse than this combination of techniques.
>
> 1) We discuss the choice of using likelihood(ratios) over posterior estimation (SNPE) and show empirical results in Sec 4.3.
> 2) We show that a mode-seeking divergence (Wiqvist et al. 2021) works less well than any of the three objectives we propose. In addition, we will add a supp figure that demonstrates that the choices of self-normalized weights and using the STL are required for performance.
> 3) For SIR, we show that it consistently improves performance. There might be other importance-sampling (IS) schemes that might also work (we investigate multiple variational objectives, but we did not investigate multiple IS schemes).
>
> > The method without SIR is not sufficient ensure the quality of the posterior samples despite expressiveness of NFs (as well as how the SIR step works)
>
> The comparison of the method with and without SIR is shown in Figure 3 (dotted vs solid lines). However, we acknowledge that we have neither explained SIR in detail, nor demonstrated why it works. We will add a paragraph and figure that show that SIR fixes inaccuracies of the learned variational posterior (even with NFs), and show why it is particularly useful for mass-covering divergences. We will also outline SIR in pseudocode.
>
> > The calibration kernel fix and the extra bias-factor model is not arbitrary and its learning or selection does not bear too much additional computational hurdle.
>
> The calibration-kernel is used to deal with the ubiquitous problem that scientific simulators, when fed with unrealistic parameters, often return invalid outputs when a simulator does not run through (e.g. NaN). In that sense, it is a technical solution to a nuisance faced by all likelihood(-ratio) based approaches (SNLE, SNRE), not just SNVI. These invalid simulations are not useful for accurately learning the likelihood, and we would like to ignore them. This strategy has been applied in **posterior** estimation approaches, but **likelihood**(-ratio)-based approaches had been lacking this feature.
> We 1) derive the bias-factor that emerges when ignoring invalid simulations in likelihood(-ratio)-based approaches and 2) devise a strategy to learn this bias-factor. The bias-factor is not arbitrary, but it naturally arises depending on the ratio of invalid simulations to valid ones. Fig4 demonstrates that the bias factor can be learned robustly. We will add an additional figure that shows that the comp. cost of estimating the bias factor is small compared to the likelihood. We realize that we did not do a great job at explaining the intuition behind the calibration kernel, and will explain it more clearly in the manuscript.

---

> > ### Comment · Reviewer_NhK3 · 2021-11-20
> > **Thanks and raised score**
> >
> > Thank you for your response and clarifications, and especially on acting on the feedback.
> >
> > The authors has addressed my concerns in detail and communicated how they would revise the paper accordingly for the camera-ready version. I would lean towards acceptance.

---

### Author Response · Authors · 2021-11-13
**Overall response to reviews**

We thank the reviewers for their constructive feedback, insightful comments and helpful suggestions, and their generally positive appraisal of our work. We appreciate that all reviewers recognize that SNVI “improves efficiency” (R 9Rvr) relative to existing methods (R JNHt) and their positive comments regarding the presentation of the paper (R NhK3, NxCC, 9Rvr). In addition, the reviewers acknowledge that the “results are good” (R NxCC, largely outperforming the previous state-of-the-art, e.g. on the pyloric network model, Section 4.3) and that we performed a “relatively complete ablation study” (R 9Rvr).

We emphasize that SNVI leads to substantial gains in computational performance and simulation-efficiency over existing neural likelihood (and likelihood-ratio) methods for simulation-based inference over a range of tasks, and that we expect it to be the ‘go-to’ method for simulation-based inference. We do hope that our modifications convince the reviewers to increase their scores.

We here briefly address concerns shared across multiple reviewers-- we appreciate their comments and believe that addressing them will considerably strengthen the paper. In particular, we will include new supplementary figures that address several concerns:

Concern 1: The work is a combination of existing techniques
R NhK3  and R 9Rvr expressed concerns that SNVI is based on a combination of previous techniques (R NhK3 writes that “The main weakness of the proposed approach is that it takes the form of combining multiple existing techniques into one coherent approach.”) We agree that our method is made up of several components most of which (seen by themselves) have been described in previous work. However, we do not consider this a weakness of our approach: One of our contributions is to identify issues in current SOTA methods and to select and combine components that overcome these limitations. By making specific choices about each component, we achieve a method that clearly outperforms SOTA, sometimes by orders of magnitude (Fig4). In addition, we believe that placing previous methods in context and illuminating their potential for use in SBI is an important contribution of the paper in itself.

Concern 2: Insufficient explanation of the importance and the use-cases of the calibration kernel
We agree that the calibration kernel has been insufficiently explained in the submission, and will improve it in the paper. The calibration-kernel is used to deal with the (annoying but unfortunately ubiquitous) problem that scientific simulators, when fed with unrealistic parameters, often return invalid outputs (e.g. fail to run and output NaN). To clarify the necessity of the calibration kernel, as well as the exact workings of the calibration kernel, we will adapt the corresponding sections in the paper (details in individual responses).

Concern 3: Missing comparisons to other methods
R NxCC and 9Rvr suggest a comparison to directly learning the posterior from samples. This approach is, in fact, what is known as (Sequential) Neural Posterior Estimation (SNPE) in the literature. We describe SNPE (including its issues) in the paper. We compare our method to (S)NPE in Section 4.3, and find that its issues prohibit the use of SNPE on the studied model (as would be the case for many other models of interest). Furthermore, we point out that our empirical comparisons and metrics are based on benchmark tasks for which results for many other methods are available (Lueckmann et al 2021) -- thus, one can directly compare performance to other (and future) methods. We will clarify this point in the paper.

Concern 4: Missing guidance on which metric to use
R NhK3 and R NxCC are missing a recommendation for which of the three introduced optimization objectives for SNVI to use. While all three methods show highly similar performance on the problems we looked at (all outperforming other methods), we realize that such a recommendation is desirable for readers and users. We will add a paragraph in the paper that discusses different aspects of each method and that will allow users to make an informed choice for the specific problem at hand (taking into account computation time of the simulator and architecture of the normalizing flow).

We are addressing all of these concerns in more detail in the individual responses below. We hope that this response convinces you that some of your concerns are already addressed in the paper (e.g. comparison to SNPE), and that the suggested changes to the paper provide insights into the usefulness of SNVI on many real-world problems.

Thank you for taking your time and reviewing our paper!
The authors

---

### Author Response · Authors · 2021-11-22
**Uploaded revised version**

We thank all reviewers for engaging with our paper and providing helpful feedback which substantially strengthened our submission. We have just updated the manuscript with the promised changes. Thank you!

Best wishes
The authors

---

### Public Comment · ~Michael_Deistler1 · 2022-10-24
**Correction**

We have been pointed to an issue in one of the results on the pyloric network (Fig. 4). We have updated the [arxiv version of the paper](https://arxiv.org/abs/2203.04176) to describe and correct this issue, and have updated the code in the [GitHub repository](https://github.com/mackelab/snvi_repo).

Best wishes, the authors

---

### Decision · Program_Chairs · 2022-01-20

**Decision:**

Accept (Spotlight)

**Comment:**

The authors propose a well-presented approach to likelihood-free inference. The reviewers are all in alignment in recommending this paper for acceptance. There was a healthy discussion between authors and reviewers, where the authors have already incorporated many of their recommendations. The potential for this methodology to be applied to situations with expensive simulators should be intriguing to a broad audience. As a result, I recommend for this paper to be accepted as a spotlight.